# Optomechanical ring resonator for efficient microwave-optical frequency conversion

I-Tung Chen[1], Bingzhao Li[1], Seokhyeong Lee[1], Srivatsa Chakravarthi[2], Kai-Mei Fu [1,2,3] & Mo Li [1,2] ✉

Phonons traveling in solid-state devices are emerging as a universal excitation for coupling different physical systems. Phonons at microwave frequencies have a similar wavelength to optical photons in solids, enabling optomechanical microwave-optical transduction of classical and quantum signals. It becomes conceivable to build optomechanical integrated circuits (OMIC) that guide both photons and phonons and interconnect photonic and phononic devices. Here, we demonstrate an OMIC including an optomechanical ring resonator (OMR), where co-resonant infrared photons and GHz phonons induce significantly enhanced interconversion. The platform is hybrid, using wide bandgap semiconductor gallium phosphide (GaP) for waveguiding and piezoelectric zinc oxide (ZnO) for phonon generation. The OMR features photonic and phononic quality factors of $>1 \times 10^5$ and $3.2 \times 10^3$, respectively. The optomechanical interconversion between photonic modes achieved an internal conversion efficiency $\eta_i = (2.1 \pm 0.1)\%$ and a total device efficiency $\eta_{tot} = 0.57 \times 10^{-6}$ at a low acoustic pump power of 1.6 mW. The efficient conversion in OMICs enables microwave-optical transduction for quantum information and microwave photonics applications.

Although photonic integrated circuits have become a mature technology, the development of phononic integrated circuits is still at an early stage[1–5]. The photonic ring resonator is a crucial component in photonic integrated circuits, as it allows for efficient electro-optic modulation, frequency comb generation, wavelength filtering, routing, and switching[6–14]. Similarly, phononic ring resonators are expected to play an important role in phononic circuits and have been recently demonstrated in various materials, such as lithium niobate and gallium nitride (GaN) on sapphire substrates[2,4,5] and GaN on silicon carbide[15]. A ring resonator for both photons and phonons—an optomechanical ring resonator (OMR)—would resonantly enhance the interaction between co-circulating photons and phonons to achieve ultrahigh optomechanical coupling efficiency, thus highly desirable for the abovementioned applications. Wavelength scale optomechanical waveguide that confines both photons and phonons for optomechanical conversion has recently been demonstrated[16,17]. However, realizing OMRs with high-quality factors requires low-loss waveguides for

both photons and phonons, as well as a phonon source that can generate phonons in resonance with the ring. As macroscale acoustic and optical co-resonator has been demonstrated using high-overtone bulk acoustic resonators (HBAR)[18], integrated OMR with photonic and phononic mode in co-resonance has yet to be demonstrated. Therefore, its realization faces challenges in material selection, structure design, and device fabrication and thus remains elusive to date. Such OMRs interconnected in optomechanical integrated circuits (OMIC) would offer a wide range of applications in both classical and quantum domains. These applications include quantum transduction[19–25], single-photon gate operations[26], nuclear spin control[27], mechanical control of defect centers[28–30], frequency comb generation and modulation[31], frequency conversion[17,32,33], and non-reciprocal optical devices[34–37].

Here, we demonstrate an OMR built on a hybrid material platform, on which non-piezoelectric material gallium phosphide (GaP) is used as the optomechanical photon-phonon guiding layer and combined with piezoelectric material zinc oxide (ZnO), which is used to

[1]Department of Electrical and Computer Engineering, University of Washington, Seattle, WA 98115, USA. [2]Department of Physics, University of Washington, Seattle, WA 98115, USA. [3]Physical Sciences Division, Pacific Northwest National Laboratory, Richland, Washington 99352, USA. ✉e-mail: moli96@uw.edu

electromechanically generate phonons. The hybrid approach allows us to exploit the best properties of both materials: GaP provides a high optical refractive index[38–40] ($n = 3.1$ at 1.55 μm wavelength), a large bandgap[41] (2.26 eV), a high $\chi^{(2)}$ nonlinearity[42–45], and a high opto-mechanical figure of merit[46–48], while ZnO possesses strong piezo-electricity and can be easily sputtered as a thin film[30,49]. Moreover, boron-doped GaP can be grown epitaxially on silicon wafers[50] by commercial vendors (see "Methods" and Supplementary Note 6), enabling a scalable platform to build large-scale optomechanical integrated circuits (OMICs). To enable waveguiding of both photons and phonons, the GaP layer must be suspended from the silicon sub-strate. This configuration allows us to achieve high-quality factors and high optomechanical coupling efficiency, making the ZnO/GaP hybrid a promising platform.

## Results

### The device design and simulations

Figure 1a depicts our design of OMR, which is coupled with pairs of photonic and phononic waveguides that enable independent coupling of photons and phonons, respectively, in and out of the OMR. It is important to distinguish the OMR from earlier cavity optomechanical and integrated acousto-optic devices. In cavity optomechanical sys-tems such as optomechanical crystals[51–53], localized photonic and

phononic modes are coupled. In contrast, OMR features co-resonating itinerant photons and phonons[54,55]. When the co-resonating and phase-matching conditions (PMC) are satisfied by the photonic and phononic modes, their interaction is resonantly enhanced. The co-circulating configuration also contrasts with earlier integrated acousto-optic sys-tems, where phonons interact with the photonic mode in one or a few passes only[34,56,57].

We consider the interaction between two photonic modes (mode 0 and 2) and one phononic mode in the OMR with angular frequencies and wavenumbers of $(\omega_0, \beta_0)$, $(\omega_2, \beta_2)$, and $(\Omega, \kappa)$, respectively. The PMC for optomechanical mode conversion through the Brillouin process[58] is $(\omega_2, \beta_2) = (\omega_0 \pm \Omega, \beta_0 \pm \kappa)$. The plus and minus signs are determined by whether the scattering is an anti-Stokes or Stokes process. The optomechanically induced mode conversion process follows the coupled-mode theory (CMT) as:

$$\frac{d}{dz}A_2(z) = -iG_{20}A_0(z)e^{-i\Delta\beta} \tag{1}$$

where $\Delta\beta$ is the phase-mismatch. $G_{20}$ is the total optomechanical coupling coefficient that converts the energy from photonic mode 0 to mode 2 and is proportional to the square root of phonon flux $\Phi = P_a/\hbar\Omega$. $P_a$ is the acoustic power. We thus define flux-normalized

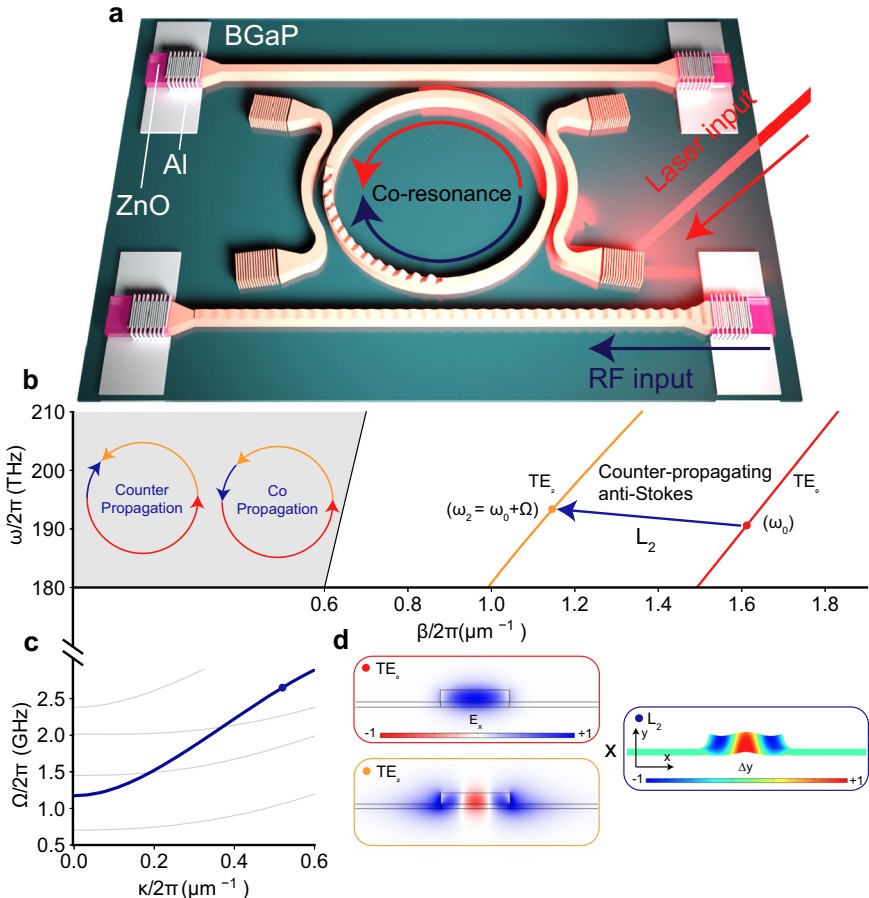

**Fig. 1 | Optomechanical ring resonator (OMR) with co-resonating photon and phonon modes. a** Schematic illustration of the optomechanical integrated circuit (OMIC) device, which consists of photonic and phononic waveguides coupled with an OMR. **b** The photonic dispersion curve of the OMR waveguide, which supports TE$_0$ (red) and TE$_2$ (orange) modes. The counter-propagating TE$_0$ ($\omega_0$) photonic mode and the L$_2$ ($\Omega$) phononic mode (blue arrow) are phase-matched to generate the TE$_2$ ($\omega_2 = \omega_0 + \Omega$) mode through the anti-Stokes scattering process. The red,

yellow, and blue arrows represent the TE$_0$, TE$_2$ and the L$_2$ mode traveling directions inside the OMR. **c** The phononic dispersion of the OMR waveguide. The L$_2$ mode is denoted as the solid blue line. **d** The cross-sectional mode profiles of the TE$_0$, TE$_2$, and the L$_2$ modes. The color bar of the L$_2$ mode represents the y-component displacement field. The color bar of the TE$_0$ mode represents the x-component of the electric field.

coupling coefficient $g_{20}$, which is derived from perturbation theory and given by

$$g_{20} \equiv \frac{G_{20}}{\sqrt{\Phi}} = -\frac{\omega_2}{2} \frac{\int dA \mathbf{E}_2^* \cdot \delta\varepsilon \cdot \mathbf{u}(x,y) \cdot \mathbf{E}_0}{P_2 \sqrt{\Phi}} \tag{2}$$

Here $A_0$ and $A_2$ are the electric field amplitude of two photonic modes; $\mathbf{E}_0$ and $\mathbf{E}_2$ are the electric fields, $\mathbf{u}$ is the mechanical displacement field inside a resonator, which induces permittivity perturbation $\delta\varepsilon$, and $P_i$ is the optical power in mode $i$ (See Supplementary Note 1).

To achieve both phase-matching and co-resonating conditions, we design the OMR with a rib waveguide that supports $TE_0$ and $TE_2$ photonic modes. Figure 1b shows the simulated optical dispersion curve of the $TE_0$ and $TE_2$ modes of a rib waveguide with a width of 1.01 µm and a rib height of 150 nm. The waveguide is also a phononic waveguide, supporting a second-order Lamb ($L_2$) mode with the dispersion curve shown in Fig. 1c. It is possible to find an OMR diameter such that the $TE_0$ and the $L_2$ modes are co-resonating at a desired wavelength and frequency. From the dispersion relations, we find the PMC is satisfied at a point as marked in Fig. 1b. At -193.55 THz (vacuum wavelength 1548.91 nm), the $TE_0/TE_2$ wavenumber difference $\Delta\beta/2\pi = 0.498\ \mu m^{-1}$ and frequency difference $\omega_2 - \omega_0$ (Fig. 1b) match the dispersion of the $L_2$ mode, yielding a phonon frequency of $\Omega/2\pi = (\omega_2 - \omega_0)/2\pi = 2.56$ GHz (Fig. 1c). We then can determine the OMR diameter to be 200 µm for the $TE_0$ and $L_2$ modes to be co-resonating. The symmetry of the $TE_0$, $TE_2$, and $L_2$ modes, as shown in Fig. 1d, ensures that the mode overlapping integral in (2) is non-vanishing. As a result, both phase-matching and co-resonating conditions are satisfied, leading to very efficient optomechanical mode conversion between the three modes.

## A hybrid piezo-optomechanical material platform

We fabricate the OMIC on a commercial boron-doped GaP-on-Si wafer with a 266 nm GaP layer (See "Methods" and Supplementary Note 6). The bottom layer of Fig. 2a shows an optical microscope image of the completed device, which contains three main elements: the OMR in the center, two pairs of photonic waveguides with grating couplers,

and two pairs of phononic waveguides. The configuration of the photonic and phononic circuitries is similar to an add-drop filter, as highlighted in blue and red in the middle and top layers, respectively, in Fig. 2a. To generate and detect the acoustic waves, on each end of the phononic waveguide, an interdigital transducer (IDT) with a period $\Lambda = 2\ \mu m$ is fabricated on a layer of ZnO that is sputtered and patterned. ZnO's strong piezoelectricity is used to efficiently generate the acoustic waves in our experiment, although GaP has a non-zero piezoelectric coupling coefficient, it is too weak for us to measure RF resonance and transmission using only GaP. The two identical phononic waveguides start with a width of $W = 15.0\ \mu m$ in the IDT region to achieve optimal impedance matching with the RF source, and linearly taper to a width of $W = 1.0\ \mu m$. The silicon under the GaP is undercut to suspend the whole OMR device in air to minimize the phononic loss through the substrate. The phononic waveguide then evanescently couples to the OMR through the partially etched GaP slab in a pulley configuration with a gap $g_a = 200$ nm and length of $l_a = 13\ \mu m$. The two photonic waveguides couple to the OMR also with a pulley configuration but have different designs to selectively couple the $TE_0$ and $TE_2$ modes. One optical coupler is designed to in-couple $TE_0$ mode into the OMR while the other is designed to out-couple the $TE_2$ mode from the OMR and convert it to the $TE_0$ mode for readout through the grating couplers. Detailed design parameters are included in Supplementary Table S2.

## OMR phononic and photonic characterization

The separated photonic and phononic circuitries allow us to characterize the photonic and phononic properties independently before we combine them to demonstrate optomechanical conversion. The phononic properties are probed by exciting the acoustic wave electromechanically using an RF source to actuate the IDT at one end of the waveguide. The acoustic wave will propagate along the waveguide, couple into the OMR, and exit at the IDTs on the other end of the waveguide. Similarly, the photonic properties are characterized using laser sources and fibers coupled with the grating couplers. The symmetric design of the two pairs of photonic and phononic waveguides allows us to reverse the propagation directions of photons and phonons independently by exchanging the input and output ports,

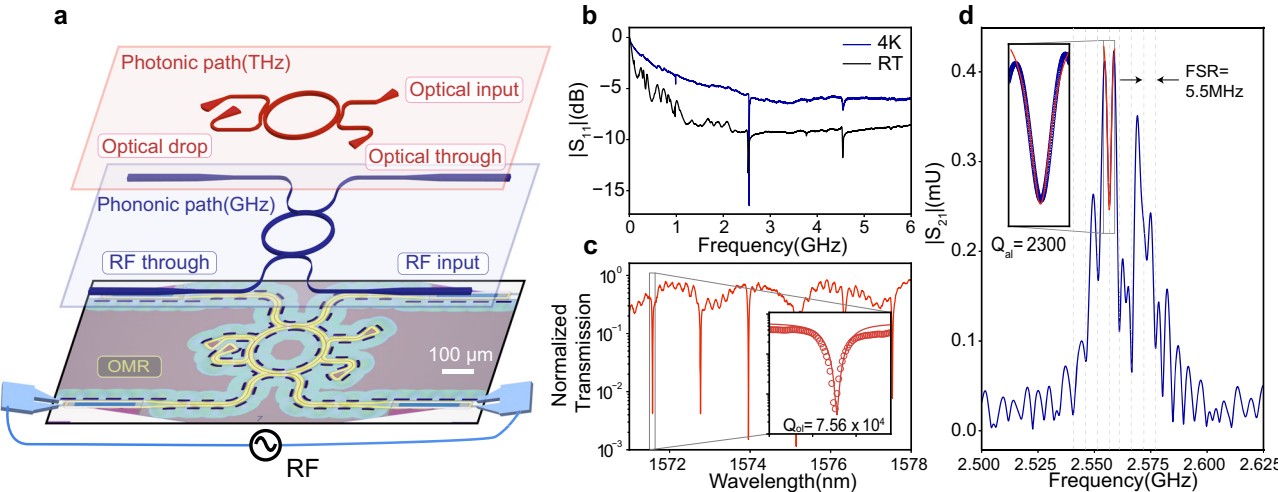

**Fig. 2 | Phononic and photonic resonances of the optomechanical ring (OMR). a** The bottom layer is an optical image of the optomechanical integrated circuit (OMIC) device. Scale bar: 100 µm. The middle and the top layers are schematic illustrations of the phononic and photonic circuitry, respectively, of the OMIC. **b** Broadband $|S_{11}|$ spectrum of one of the IDTs measured at 4 K (blue) and room temperature (black). **c** Normalized optical transmission spectrum of the OMR. The inset shows the zoomed-in resonance at 1571.7 nm, the red line is the Lorentzian

fitting of the resonance, and the open circles are the measured data. The loaded optical quality factor $Q_{ol} = 7.5 \times 10^4$. **d** Time-gated transmission spectrum $|S_{21}|$ of the OMR measured at 4 K. The spacing of the gray dashed lines denote the FSR (5.5 MHz) of the phononic resonances around 2.56 GHz. The inset shows the zoomed-in $|S_{21}|$ at 2.56 GHz, the red line is the Lorentzian fitting of the resonance, and the open circles are the measured data. The loaded acoustic quality factor $Q_{al} = 2300$.

enabling us to test the time-reversal symmetry and reciprocity of the device.

We first characterize the device's phononic properties. Figure 2b shows the broadband RF reflection spectrum ($S_{11}$) measured at one IDT using a vector network analyzer (VNA). The measurement is conducted at room temperature (RT) and at low temperatures using a cryogenic probe station with a stage temperature of 4 K. Several pronounced resonance peaks can be observed and correspond to the excitation of phononic modes. Based on our simulation and design, the resonance at ~2.56 GHz is the $L_2$ mode of interest for optomechanical conversion. The resonance signal is stronger at low temperatures, indicating higher electromechanical conversion efficiency, which we attribute to the lower serial resistance of the aluminum IDTs and the resultant better impedance matching. Fitting the spectrum with the Butterworth Van Dyke (BVD) model[59,60] yields electromechanical conversion efficiency of 90% at 4 K, compared to 25% at room temperature (see Supplementary Note 4).

We then measure the transmission coefficient ($S_{21}$) of the phononic modes between the two IDTs on the two ends of the phononic waveguide. To remove the parasitic coupling of RF signals through the free space and acoustic wave reflections within the IDTs, we use the time-gating filter on the VNA to remove the signal <190 ns time delay, because the expected phononic transmission time between the pair of IDTs is 250 ns (See "Methods" and Supplementary Note 4). The measurement result at 4 K is shown in Fig. 2d. The spectrum shows a main transmission window centered at 2.56 GHz with a −3dB bandwidth of 25 MHz, which is determined by the IDT design. Within this bandwidth, a series of evenly spaced peaks can be observed. These come from the

OMR's phononic resonances with a free-spectral range (FSR) of 5.5 MHz. The corresponding phonon group velocities are 3450 m/s, which is consistent with the simulated results for $L_2$ mode. The intrinsic quality factor from the phononic resonance is calculated to be $Q_{ai} = 3.2 \times 10^3$, corresponding to a phonon propagation loss of $\alpha_a = 6.2$ dB/mm. We have identified material defects at the interface between silicon and GaP in our material system[61]. We suspect these defects are the dominant phononic loss channel that is temperature-independent. As a result, the improvement in the phononic Q-factor at cryogenic temperatures is not significant. The optical transmission spectrum of the device measured through the $TE_0$ waveguide is displayed in Fig. 2c, showing $TE_0$ mode resonances with an intrinsic quality factor of $Q_{oi} = 1.5 \times 10^5$. These results show that the OMR has simultaneous phononic and photonic resonances with reasonably high-quality factors. In addition, the waveguide coupling scheme works for both types of waves on an integrated device platform.

### Microwave-optical optomechanical conversion experiments

We next explore the optomechanical conversion process in the OMR by inputting both photons and phonons and tuning them to co-resonating. As there are two pairs of bidirectional waveguides and couplers, we have the freedom to choose the propagation directions between the acoustic wave and the optical wave. We can configure them to be either co-propagating or counter-propagating, allowing us to investigate the PMCs. Figure 3 illustrates the various combinations of photon and phonon propagation directions, which includes one axis for photon propagation directions and one for phonon propagation directions. The first and the third quadratures correspond to co-

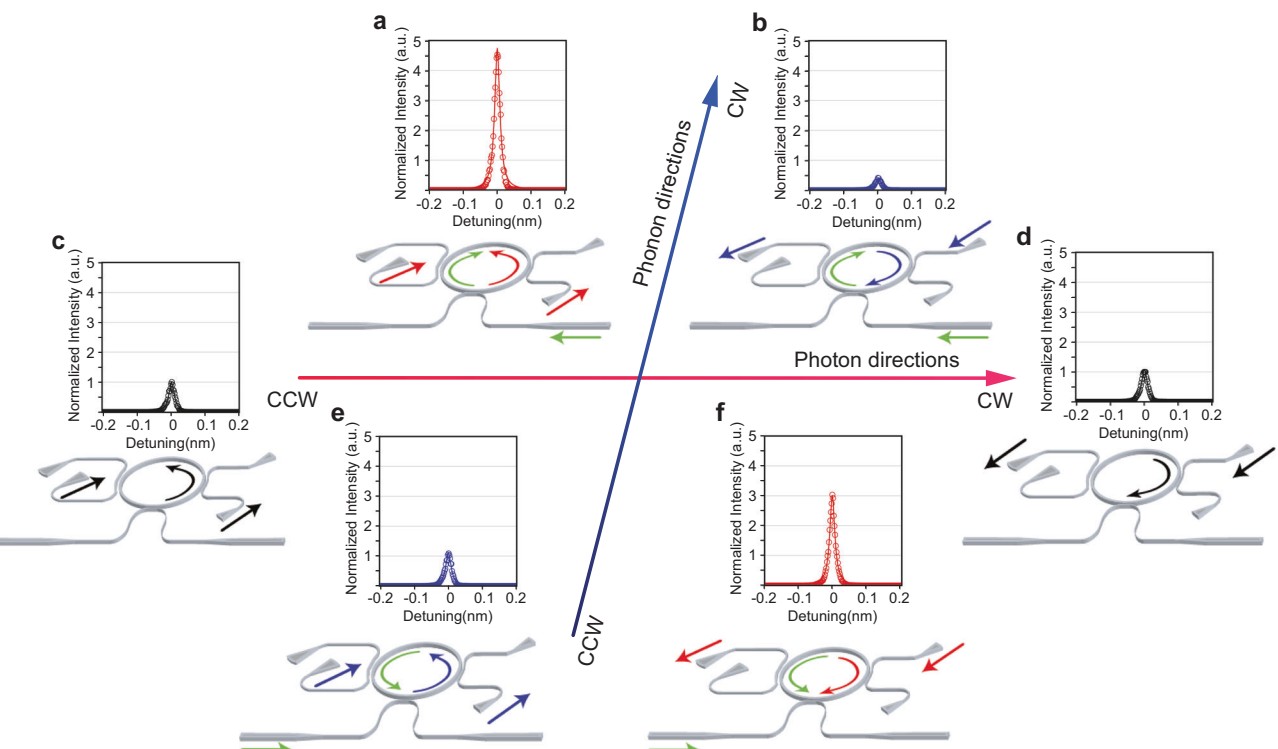

**Fig. 3 | Time-reversal symmetry and optical non-reciprocity of the opto-mechanical ring resonator (OMR).** The OMR is measured with different configurations in the four quadratures made of photon and phonon propagation directions. The $TE_2$ output signal is measured and plotted as the normalized intensity in arbitrary units (a.u.). **a** and **f** are the cases where the photon and phonon are counter-propagating, thus in the second and fourth quadrature, respectively. They are time-reversal of each other. **b** and **e** are cases where the photon and phonon are co-propagating, thus in the first and third quadrature, respectively.

They are also time-reversal of each other. **c** and **d** are cases when only photons circulate in the OMR with no phonons, thus on the x-axis. (**a** and **b**), (**e** and **f**) are the optical reciprocal of each other. The non-reciprocal output is due to the symmetry-breaking by phonons. The device schematics in each quadrature illustrate the selected input port for the specific propagation directions. The $TE_2$ transmission is normalized to cases (**c**) and (**d**). The solid lines are Lorentzian fitting, and the open circles are measured data.

propagating while the second and the fourth correspond to counter-propagating photons and phonons. The measurement results with both the acoustic wave (2.56 GHz) and the optical wave (1571.7 nm) tuned to the resonance of the OMR are shown in Fig. 3. The out-coupler is designed to output $TE_2$ mode only, which should be generated by optomechanical conversion at PMCs. However, even when the acoustic wave is turned off, there is some output no matter which direction the optical input is, as shown in Fig. 3c, d. We attribute the generation of the $TE_2$ without acoustic waves to the undesired intermodal scattering in the multimode photonic waveguide of the OMR and the region of optical couplers. This intermodal scattering generates a static $TE_2$ mode, which has no frequency shift and is independent of the acoustic wave power, as confirmed by optical spectral measurements. We then inject the phonons in a direction such that it is counter-propagating to the photons in the OMR, as shown in Fig. 3a, f. According to Fig. 1, the PMC is satisfied in this configuration for optomechanical conversion from the $TE_0$ mode to the $TE_2$ mode through an anti-Stokes Brillouin process. As a result, the output of the $TE_2$ mode is significantly enhanced in Fig. 3a, f, compared with Fig. 3c, d. When we reverse the direction of one of them, the PMC satisfies the co-propagating Stokes if the phonons and photons are co-propagating. However, in the co-propagating configuration, the anti-Stoke scattering is prohibited for the $TE_0$ mode but permitted from the static $TE_2$ mode (to the $TE_0$ mode, Fig. 1b), leading to the suppression of $TE_2$ mode output, as shown in Fig. 3b, e.

It is interesting to note that the freedom to change the propagation directions of both photon and phonon allows us to test the time-reversal symmetry and the reciprocity of the system. For example, the configurations in Fig. 3a, f, and b, e, respectively, are time-reversal symmetric to each other because both the photon and phonon propagations are reversed. Therefore, the output of these two configurations is equivalent because this device system has no time-reversal symmetry-breaking mechanism like a magnetic field. In contrast, the configurations in Fig. 3a, b, and e, f, respectively, are reciprocal with respect to the photonic waveguide, because the photon propagation is reversed while the phonon propagation direction is the same. Their outputs are inequivalent (Fig. 3a, b, or e, f), indicating photonic non-reciprocity. Apparently, the directional phonon propagation provides a spatial-temporal modulation of the medium that breaks the reciprocity. Such an optomechanically induced non-reciprocity has been utilized to realize non-reciprocal photonic devices such as a photonic isolator or a circulator[37,62,63].

To spectrally resolve the optomechanical conversion from $TE_0$ to $TE_2$ mode, we employ a heterodyne measurement scheme, as shown in Fig. 4a (see "Methods"). The reference signal is frequency shifted by δ using an acousto-optic frequency shifter (AOFS) to distinguish various frequency components involved. In the beating spectrum, we observe four distinct frequency components: (1) $P_{02}(\Omega - \delta)$, the beating tone between the anti-Stokes $TE_2$ mode, which is generated by the counter-propagating $L_2$ and $TE_0$ modes, and the reference signals. (2) $P_{02}(\Omega + \delta)$, the interference of Stokes $TE_2$ mode, which is generated by co-propagating $L_2$ and $TE_0$ modes, and the reference signal. (3) $P_{02}(\Omega)$, the interference between the $TE_2$ modes that is generated by static scattering without frequency shift (static $TE_2$ mode) and

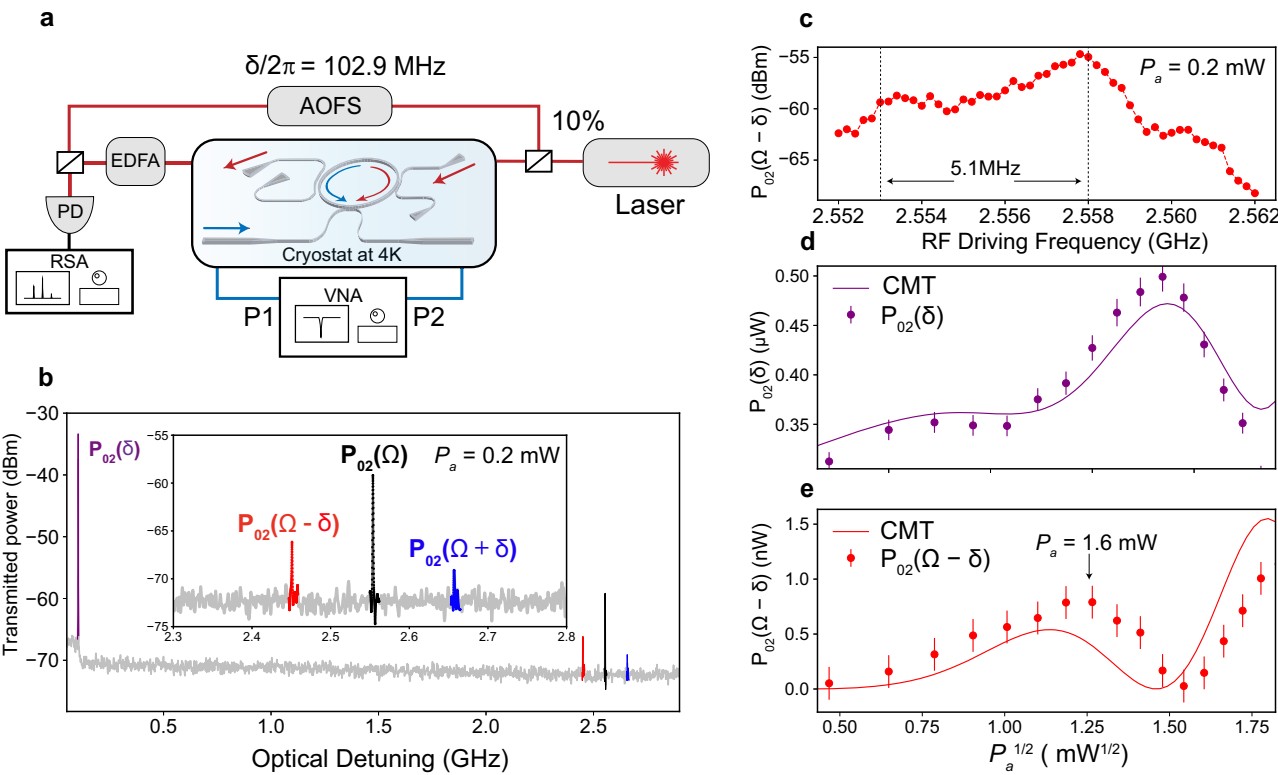

**Fig. 4 | Spectral-resolving heterodyne measurement of optomechanical mode conversion process. a** Heterodyne measurement schematic. EDFA erbium-doped fiber amplifier, RSA real-time spectrum analyzer, AOFS acousto-optic frequency shifter. The reference signal is generated by shifting the laser frequency by $\delta/2\pi = 102.9$ MHz using an AOFS. **b** The beating signals of the heterodyne measurement, the red and blue signals correspond to the counter-propagating anti-Stokes $P_{02}(\Omega - \delta)$ and co-propagating Stokes $P_{02}(\Omega + \delta)$ signals of the optomechanical ring resonator (OMR), respectively. The purple signal $P_{02}(\delta)$ is generated by the beating of the static $TE_2$ mode and reference signal. It thus provides a measure of the $TE_0$ pump intensity in the OMR. Inset: zoom-in of the signals $P_{02}(\Omega - \delta)$, $P_{02}(\Omega)$, and $P_{02}(\Omega + \delta)$. **c** Anti-Stokes signal $P_{02}(\Omega - \delta)$ as a function of acoustic wave frequency. The gray dashed line denotes the 5.1 MHz frequency separation between the two peaks. **d** $TE_0$ mode pump depletion signal $P_{02}(\delta)$ as a function of $L_2$ mode pump power. The purple solid line is the Coupled-mode theory (CMT) fitting. **e** Anti-Stokes signal $\mathbf{P_{02}(\Omega - \delta)}$ as a function $L_2$ mode pump power. The red solid line is the CMT fitting. The solid circles within the error bar in (**d** and **e**) represent the mean of the measurements.

optomechanical conversion (both anti-Stokes and Stokes). And (4) $P_{02}(\delta)$, the interference between the static $TE_2$ mode and the reference signal. Because the static $TE_2$ mode power is proportional to the $TE_0$ mode power circulating in the OMR, it provides a measurement of the pump power.

Ideally, the counter-propagating photons and phonons are phase-matched to produce only the anti-Stokes signal $P_{02}(\Omega - \delta)$ inside the OMR. However, we also observe the Stokes signal $P_{02}(\Omega + \delta)$. We attribute the observed Stokes signal to the back-reflected $L_2$ mode that is co-propagating with $TE_0$ mode. The co-propagating $L_2$ and $TE_0$ modes satisfy the PMC by annihilating a $TE_0$ photon and creating a Stokes-shifted $TE_2$ photon and an $L_2$ phonon. The asymmetry (3 dB differences) between $P_{02}(\Omega - \delta)$ and $P_{02}(\Omega + \delta)$ in Fig. 4b confirms that the counter-propagating anti-Stokes process dominates, which is consistent with the input configuration.

To further characterize the PMC around resonance condition, we change the RF pumping frequency within the IDT bandwidth and measure the dependency of $P_{02}(\Omega - \delta)$ on $L_2$ frequency. We expect the peak frequencies of $P_{02}(\Omega - \delta)$ to match the phononic resonances of the OMR as in Fig. 2d. The results in Fig. 4c indeed show two peaks of $P_{02}(\Omega - \delta)$ spaced by 5.1 MHz within the IDT bandwidth (10 MHz at 4 K). This spacing, however, deviates from the phononic resonance FSR (5.5 MHz). We attribute this mismatch to the frequency misalignment between the resonance condition of the $L_2$ mode and the PMCs as calculated in Fig. 1b.

We next monitor the beating signals by varying the acoustic power to characterize mode conversion and determine the optomechanical coupling coefficient $g$. While the $P_{02}(\Omega - \delta)$ signal directly measures the generated $TE_2$ mode, the $P_{02}(\delta)$ signal provides a measure of the $TE_0$ pump power in the OMR. The internal conversion efficiency $\eta_i = P_{TE0}/P_{TE2}$ is calculated by converting $P_{02}(\delta)$ and $P_{02}(\Omega - \delta)$ to $P_{TE0}$ and $P_{TE2}$ after considering the photoreceiver gain, EDFA gain, and reference signal power. Figure 4d, e shows the measured $P_{02}(\Omega - \delta)$ and $P_{02}(\delta)$ signals as a function of the acoustic power $P_a$. We observe a clear oscillation in $P_{02}(\Omega - \delta)$ and $P_{02}(\delta)$, which behaves differently from conventional mode converters and needs to be modeled with the CMT in a ring resonator. Without additional acoustic power-dependent loss in the CMT, we can solve Eq. (1) and the amplitude of the $TE_2$ and $TE_0$ modes can be expressed as

$$A_0(z) = A_0(0)\cos(Gz) \tag{3}$$

$$A_2(z) = -iA_0(0)\sin(Gz) \tag{4}$$

where $A_0(A_2)$ is the amplitude of $TE_0$ ($TE_2$) mode, $z$ is the interaction length between the two modes. However, this simple model cannot fully represent our measured data, so we incorporate the addition loss channel and a phase-mismatch to our model. Particularly, the conversion to the $TE_2$ mode can be considered as an additional loss to the $TE_0$ mode pump, which we model with an acoustic power-dependent loss rate $\gamma(P_a)$ (See Supplementary Note 1). $\gamma(P_a)$ alters the waveguide to OMR coupling condition and thus the circulating pump power, complicating the situation. Our model captures these effects and fits the data with good agreement, as shown in Fig. 4d, e.

The full parameter set that is used for fitting is listed in Supplementary Table S1. From the fitting, we extract the coupling coefficient $\frac{g}{\sqrt{\hbar\Omega}} = 230\ mm^{-1}\sqrt{W^{-1}}$ and the phase-mismatch $\Delta\beta = 0.08\ \mu m^{-1}$, respectively. The achieved highest internal conversion efficiency $\eta_i \equiv \frac{P_{TE_2}}{P_{TE_0}} = (2.1 \pm 0.1)\%$ at $P_a = 1.6$ mW, which agrees with the theory (See Supplementary Note 2). The phase-mismatch can be mitigated by tuning the resonance frequency of OMR using photothermal tuning[64]. We can calculate the critical acoustic power needed to achieve unity conversion efficiency in the OMR at phase-matching condition:

$$P_{\pi/2} = \left(\frac{\sqrt{\hbar\Omega}}{2gD}\right)^2 = 0.1\ mW$$

where $D = 200\ \mu m$ is the diameter of the OMR. The $P_{\pi/2} = 0.1$ mW is projected at phase-matching condition for unity conversion efficiency, however, in our experiment, we can only achieve $P_{\pi/2} = 1.6$ mW with a total conversion efficiency $\eta_{tot} \equiv \frac{P_{out}}{P_{laser}} = 0.57 \times 10^{-6}$.

## Discussion

In conclusion, we have demonstrated the first OMR in which photonic and phononic modes are co-resonantly coupled to achieve efficient optomechanical mode conversion. The integration of the OMR in an OMIC allows exploration and validation of the time-reversal symmetry and optical reciprocity of the system. The OMR is a new type of traveling wave cavity optomechanics, which has unique advantages over the standing-wave type, including multimode coupling, readiness for interconnecting multiple devices, and flexibility in device design. The critical next step is to improve the photonic and phononic resonance quality factors, which can be achieved with fabrication process optimization and new material platforms. It is also necessary to develop an unsuspended design where the waveguiding material and the substrate have both high refractive index contrast and large acoustic velocity differences. The hybrid design of the OMIC is applicable to the integration of many piezoelectric and non-piezoelectric materials, such as aluminum nitride and silicon, which are CMOS-compatible. The OMR is compelling for classical applications such as high-efficiency acousto-optic frequency shifters, frequency beam splitters, and on-chip optical isolators. For quantum applications, achieving phase-matching and co-resonating in the OMR will lead to ultra-efficient microwave-to-optical signal transduction for superconducting qubits. Furthermore, the hybrid OMIC can also be integrated with many other types of solid-state qubits[65] to use phonons as quantum information carriers.

## Methods

### Device fabrication

The as-grown boron-doped GaP (266 nm)-on-Si substrate is purchased from NAsPIII/V GmbH. The GaP photonic and phononic waveguides are patterned with electron beam lithography (EBL) (JEOL-JBX6300FS) using PMMA 950 K A6 resist. The pattern is transferred into GaP by partial etching in an inductively coupled plasma etcher using chlorine-based chemistry. The 290 nm thick ZnO film is deposited using an RF magnetron sputtering system and lift-off in a sonicated acetone bath. The IDT pattern is written using the EBL and lift-off after depositing 220 nm aluminum film using an E-beam evaporator. The releasing vias are patterned by the photolithography and etched in ICP. Finally, $XeF_2$ etching of silicon is used for suspending the GaP layer. See Supplementary Note 6 and Supplementary Fig. S7 for fabrication process flow.

### Device measurement

The acoustic transmission spectrum ($S_{21}$) shown in Fig. 2 is measured by VNA (Keysight N5230C PNA-L) on a cryogenic probe station (Lakeshore CRX-4K). Prior to the $S_{21}$ measurements, the VNA was calibrated at room temperature and under vacuum. We calibrate the VNA to the tips of the RF probes as the reference using a calibration substrate (GGB Inc. CS-15).

The optical measurement scheme used in Fig. 3 is shown in Supplementary Note 5. The sample is measured in the same cryogenic probe station as in Fig. 2. The laser wavelength (Santec TSL-710) is fixed at 1571.67 nm while the RF drive is set on resonance at 2.56 GHz by the VNA. The input laser is routed to the chip via fiber probes, the output

signal is received and analyzed with an optical spectrum analyzer (Yokogawa AQ6374). The heterodyne measurement set-up shown in Fig. 4a is used to resolve the frequency shift of the mode scattering process. An acousto-optic frequency shifter (AOFS) (Brimrose model AMF-100-1550-2FP) is used to generate a reference signal with a frequency shift of 102.9 MHz. An erbium-doped fiber amplifier (EDFA) (PriTel LNHP-PMFA-23) is used to amplify the signal before a photodetector (Thorlabs RXM25AF). The photodetector output is analyzed with a real-time spectrum analyzer (Tektronix RSA5100B).

## Data availability

The data underlying the figures of this study are available at https://doi.org/10.5281/zenodo.10012019. All raw data generated during the current study are available from the corresponding authors upon request.

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

## Acknowledgements

This work is supported by the National Science Foundation (Award No. EECS-2134345 and ITE-2134345). Part of this work was conducted at the Washington Nanofabrication Facility/Molecular Analysis Facility, a National Nanotechnology Coordinated Infrastructure (NNCI) site at the University of Washington with partial support from the National Science Foundation via awards NNCI-1542101 and NNCI-2025489.

## Author contributions

I.T.C. and M.L. designed and planned the experiment; I.T.C. fabricated the samples with the recipe developed by B.L. and S.C.; I.T.C. performed measurements, data analysis and FEM simulations with contribution from B.L., S.L., K.M.F., and M.L.; M.L. supervised the project. I.T.C. and M.L. prepared the manuscript with discussions and inputs from all authors.

## Competing interests

The authors declare no competing interests.
