## [Peer Review File · Nature Communications]

Optomechanical ring resonator for efficient microwave-optical frequency conversionREVIEWER COMMENTS

Reviewer #1 (Remarks to the Author):

The paper shows an experimental setup to convert microwaves-optical photons via photonic and phononic modes on co-resonance. This reviewer finds this setup very interesting, mainly because of the different materials in the photonic integrated circuit. However, material choices and parameters impacting conversion efficiency should be further discussed.

The paper is well written, and this reviewer supports it for publication if the design choices, measurement methods, and setup are better explained in the main text.

Detailed comments.

1. Define acoustic pump power - Is this the pump power to excite the phononic modes? What is the laser pump power required to excite the photonic modes?
2. At what temperature are the measurements performed? My understanding is that the temperature is 4K. Is there a reason why the temperature is not lower?
3. How are the materials chosen? Are there other options? Lithium niobate and gallium nitride are mentioned earlier in the manuscript.
4. What are the design tolerances? Is there any tuning mechanism to meet the phase-matching conditions? It is noted that the phase mismatch is one of the main reasons for the limited conversion efficiency. So this is essential information.
5. What are the other reasons for the limited conversion efficiency? Can it improve at lower temperatures?
6. It would be better to add some parts of the supplementary information in the main text to explain (i) the fitted method, (ii) the interaction between the modes, and (iii) how the conversion efficiency is calculated.
7. In supplementary information IV, it is noted that there is a reflection of the acoustic wave. How does this impact efficiency?
8. What is phononic resistance?
9. What is the relationship between Q_{at} and Q_{oi} ?
10. Finally, it is written, "The resonance signal is stronger at higher temperatures ... resulting in a better

impedance matching". Is this effect taken into account during the design phase?

11. The paper doesn't cover noise and added noise. As measurements were actually performed, a detailed analysis would also help understand intrinsic efficiency.

12. From the abstract. "The conversion efficiency will reach unity when ideal phase-matching conditions are achieved." We instead know that the conversion efficiency depends on several parameters, such as microwave and acoustic losses, pump power, temperature, etc. Are all these parameters considered in the analysis? Will these parameters be considered for optimization?

Reviewer #2 (Remarks to the Author):

In the manuscript entitled "Optomechanical ring resonator for efficient microwave-optical frequency conversion," the authors demonstrate a new electro-optomechanical system based that utilizes suspended GaP based waveguides to confine both optical and acoustic waves. They use ZnO electromechanical transducers to launch acoustic waves into this system, which produce resonantly enhanced inter-modal Brillouin scattering within an optomechanical ring resonator. The work has several noteworthy novel aspects surrounding the use of new materials and their use of resonantly enhanced Brillouin scattering to mediate efficient scattering. However, there are several issues that should be addressed before I can support publication.

These include:

- In reading the manuscript, it was easy to miss the fact that the waveguide is suspended. Also, motivation for the use of a suspended waveguide was either subtle or not explained. I think this is a crucial part of their system and requires a bit more motivation and explanation in the main body of the text. Perhaps just one or two more sentences would be sufficient.
- Why is it necessary use ZnO as a piezoelectric transducer material? Is the piezo coefficient of GaP too small? It would be helpful to explain/motivate this decision further in the text.
- What is the limitation of phonon losses in the current experiments? Why do you think your Q-factor didn't increase much at low temperatures? Some elaboration on this point in the main text would be helpful.
- I'm perplexed by the examination of the time reversal (TR) symmetry of the system. All classical and quantum systems exhibit time-reversal symmetry. There is no controversy here, as time reversal of any Hamiltonian yields the same behavior unless we fail to reverse time for one of the field quantities (e.g. magnetic field). Isn't it really nonreciprocity that you examine?

To reiterate:

- I think we can all agree that TR symmetry is always holds if we can apply time reversal to all dynamical variables.
- B-fields do not break time-reversal symmetry; only if we do time reversal incorrectly does one come to the incorrect conclusion.

Is there some fundamental physics you seek to discover here? Please provide further justification. Otherwise, I would recommend removing the discussion of time reversal symmetry or pushing to supplement.

- To eliminate the ambiguity of their findings, I ask the authors please indicate the demonstrated optical and electromechanical cooperativities of this system.
- The authors indicate that they use a "Flux normalized coupling coefficient," but then show a power normalized coupling on page 8. Is this a mistake?
- A 90% IDT to phonon conversion efficiency is quoted
- The definitions of conversion efficiency are very important and there is some ambiguity. What is the IDT conversion efficiency? What is the IDT power? Would the quoted 90% IDT to phonon conversion efficiency equate to a 90% percent quantum efficiency at zero Kelvin?
- I object to the manner in which the authors appear to claim a Ppi of 0.1 mW since they have not achieved it. They appear to claim a level of performance in conjunction with a speculative qualifier "if phase matching is achieved". This is highly problematic and they must adjust their claims to reflect what they have achieved. Perhaps a speculative statement regarding future performance could be included in the conclusions. However, the authors must make it clear that they have not accomplished this result. It would be more appropriate to put the Ppi that they have actually achieved. It is currently absent.
- Additionally, who uses this metric? Feels like it comes out of thin air to bolster claims. Also, what physical attributes lower P-pi in this system?
- One of the key claims of this paper needs to be amended. In particular, the following statement is not correct: "In conclusion, we have demonstrated the first OMR in which photonic and phononic modes are co-resonantly coupled to achieve efficient optomechanical mode conversion with a record low critical acoustic power ($P\pi/2$)." This paper by Yoon [Yoon, et al. *Optica* 10.1 (2023): 110-117] have previously achieved this condition through an intermodal Brillouin scattering process, and should be cited. Perhaps it would be appropriate to indicate that this is the first micro-scale system. On the other hand, the authors didn't really achieve the quoted record Ppi value or the resonance condition. So I think that this claim is not really appropriate.

In conclusion, I believe that this system is novel and interesting, however, I would need to see how the authors address the above comments and revise the claims in their manuscript before I could support publication.

Reviewer #3 (Remarks to the Author):

In 'Optomechanical ring resonator for efficient microwave-optical frequency conversion' the authors present a realisation of an efficient optomechanical conversion device, where travelling photons and phonons coexist in a ring resonator. As a result of the photoelastic and moving boundary coupling mechanisms, the confined phonons result in an inter-modal optical coupling. Through a series of optical and microwave-frequency tests the propagation-dependent phase-matching is analysed in the paper and the authors provide an outlook as to how the efficiency can be increased.

The manuscript presents an interesting solution to the challenge of efficiently converting microwave and

optical photons, and provides results interesting to researchers developing integrated optomechanical and electromechanical transduction devices. The manuscript is, on the whole, clearly written and the explanation of the coupling mechanism and the required phase-matching is outlined fully.

The use of both propagating mechanical and optical states on the chip is an important step towards scalable integrated devices capable of quantum frequency conversion, and the device the authors present is a promising realisation of this. Nonetheless, in order for me to recommend publication, a number of issues with the content of the manuscript, in particular on how the data is treated and presented and how the device is modelled need to be addressed first. I have outlined these issues and questions below.

1, In the abstract the authors state 'the optomechanical conversion between photonic modes to achieve a conversion efficiency of 8.2% at low acoustic pump power 1.6 mW.' They need to clarify this statement. Are they referring to the intrinsic device efficiency here? What would the total efficiency of the device be?

2, The authors state additionally in the abstract that 'The conversion efficiency will reach unity when ideal phase-matching is achieved.' This statement is quite vague. Again, I assume they are discussing the internal conversion efficiency inside the ring resonator. However, even in this case I find this to be a potentially overblown statement. Do they expect with the loss rates in the system that every photon in the TE0 mode will be converted to a photon in the TE2 mode? In particular, would the large coupling rate of the TE0 mode to the resonant TE2 mode be extinguished in this case?

3, The authors do not give any information as to the value of κ_a (the coupling from the phonon waveguide to the OMR in the device).

4, In the main body of the text the intrinsic Q-factor of the optical mode is listed as 6.4×10^4 , however in the figure the mode is labelled with $Q_{ol} = 4.1 \times 10^5$. Firstly, is Q_{ol} the loaded quality factor? This is not explained. If this is indeed the case - it would be unphysical to have a loaded quality factor larger than the quality factor owing to intrinsic loss. This needs to be corrected.

5, The optical transmission signal in figure 2c oscillates a lot, with the same periodicity in wavelength as for the TE0 mode. The authors should outline the source of this oscillation.

6, In the book-keeping part of section II of the supplementary information, if i take $P_{device} = 0.03 \mu W$ and normalise it by the grating coupler efficiency (0.03) and the outcoupling efficiency of the TE2 mode (0.03) I get $P_{TE2} = 33 \mu W$, rather than 22.8. There seems to potentially be a mistake here. Also the value of 3% for η_{OC} is very inconsistent with the value in table S2, where it is listed as $> 60\%$. The reasons for this mismatch should be explained. Finally - how is the outcoupling efficiency of the TE02 mode measured?

7, Similarly, if i follow the values for $\eta_{GC} = 0.03$ and $\eta'_{OC} = 0.9$ and $P_{laser} = 12e-3 W$ - i get $P_{TE0} = 324 \mu W$. With $P_{TE2} = 22.8 \mu W$ - i get an efficiency of $22.8/324 = 0.07$, rather than 0.082. It should be

noted that the coupling efficiencies are only quoted to one significant figure here, which causes a great degree of uncertainty in the device efficiency.

8, In figures 4 b and c what is the acoustic power used for these measurements?

9, In figure 4c, d and e are the powers in the y-axis the electrical powers or rather the optical powers from the device? This should be clarified. The plots appear inconsistent with a device efficiency of 0.08, as at the peak of efficiency they measure 400x more power in the $P_{02}(\Delta)$ mode than the $P_{02}(\Omega - \Delta)$ mode.

10, With regards to the microwave-frequency S21 measurement - the authors state that the electromechanical conversion efficiency is 90% in the device, and yet they need to filter the VNA measurement heavily in order to see transmitted mechanical mode. These two facts seem inconsistent. Presumably if 90% of the microwave power was being converted into travelling phonons in the waveguide, one would expect all other parasitic contributions to be much smaller. Additionally, filtering in the time domain can leave artefacts on the spectral response, and as such the authors should include the unfiltered transmission spectrum.

11, The authors simulate the mechanical and optical propagation in the device, and present a formula for calculating the optomechanical coupling g_{20} in equation 2. A full presentation of the physics of this device should contain simulations of this coupling strength from the photoelastic and moving boundary conditions they present in the first few equations in the supplementary information. What value do they simulate, and how does the value of g and $P_{\pi/2}$ they extract from their measurements compare with the simulated values?

12, The coupled-mode fitting of the data requires a large number of fixed parameters. How are the values for γ_0 , γ_{a0} , Γ and Γ_a for both the optics and the mechanical modes determined?

13, As a route to increasing the device efficiency, the authors suggest thermal tuning of the phase-mismatch, however the device needs to be operated at cryogenic temperatures. How would they realise zero phase mismatch in these conditions?

14, In the conclusion the authors state that 'we have demonstrated the first OMR in which photonic and phononic modes are co-resonantly coupled to achieve efficient optomechanical mode conversion with a record low critical acoustic power ($P_{\pi/2}$).' This is not the case from what they have presented - they have projected the number they claim to be record low and not demonstrated it. This needs to be corrected.

15, A lot of details remain missing about the potential quantum applications for an improved device, something quoted as a key motivation for this work. The authors should be much more specific as to the actual form of conversion they want to pursue with this device. They discuss the potential for ultra-efficient conversion from microwave to optical signals, however they use very large mechanical powers containing macroscopic numbers of microwave photons. They need to break down what the efficiency of a single-photon-level microwave signal conversion would be for this device in order to validate their outlook. What optical power would be required for efficient microwave-to-optical conversion?

16, Additionally - there is no mention of the noise that might be added during conversion. Discussion of these details need to be contained in an outlook concerning the quantum applications of a device.

A couple of additional minor formatting notes.

1, In the supplementary the coupled mode theory uses dimensionless parameter for the coupling in and out of the resonator. These numbers are consistently referred to as 'coupling rate' which is highly misleading, as they are dimensionless transmissivities rather than rates.

2, There seems to be some inconsistency in the labelling of the 'through port' the optical input and through are labelled clearly in figure 2a, however I believe the model of the OMR presented in the equation between equations 4 and 5 of section I of the supplementary refers to the coupling to the drop port, rather than the through port.

The Authors' Response to Reviewers' Comments

Reviewer #1:

The paper shows an experimental setup to convert microwaves-optical photons via photonic and phononic modes on co-resonance. This reviewer finds this setup very interesting, mainly because of the different materials in the photonic integrated circuit. However, material choices and parameters impacting conversion efficiency should be further discussed. The paper is well written, and this reviewer supports it for publication if the design choices, measurement methods, and setup are better explained in the main text.

Our response: We thank the reviewer for the positive and insightful comments on our work.

1. Define acoustic pump power - Is this the pump power to excite the phononic modes? What is the laser pump power required to excite the photonic modes?

Our response: The acoustic power mentioned in the main text refers to the output RF power of the vector network analyzer we used to drive the IDT. In our experiments, the power is in the range of -8 to +8 dBm RF input power.

For the optical pump, we use up to 12 dBm power range to achieve a sufficient signal-to-noise ratio. Because the grating couplers' efficiency is relatively low at -15 dB (~3%), the optical power in the device is ~-3 dBm.

2. At what temperature are the measurements performed? My understanding is that the temperature is 4K. Is there a reason why the temperature is not lower?

Our response: The main results, including all data from Fig. 3 and Fig. 4, were measured at 4 K. Only the RF reflection spectrum (S_{11}) and the optical transmission spectrum in Fig. 2 were measured at room temperature.

The experiment temperature is limited by our cryogenic probe station (Lakeshore CRX-4K)'s cooling temperature limit.

3. How are the materials chosen? Are there other options? Lithium niobate and gallium nitride are mentioned earlier in the manuscript.

Our response: We choose the materials (ZnO and GaP on Si substrate) based on the following considerations:

- 1) The material has advantageous optical and acoustic properties, including a high refractive index, low acoustic velocity, and low optical and acoustic loss, so it affords high photonic and phononic performances.
- 2) The material has a high photoelastic coefficient that can facilitate efficient photon-phonon interaction.
- 3) The material platform is scalable, preferably with commercial suppliers.
- 4) The material can be readily processed to design and fabricate high quality devices with high optical and acoustic performances.

With the above considerations, GaP stands out as an optimal optomechanical material platform. GaP has a high refractive index ($n_{\text{GaP}}=3.1$ at 1550 nm), higher than that of lithium niobate and gallium nitride ($n_{\text{LN}}=2.3$

and $n_{\text{GaN}}=2.3$ at 1550 nm). GaP also has one of the largest acousto-optic figure of merit of $M_{\text{GaP}}=44.6\times 10^{15}$ s^3kg^{-1} , which is about 1.5 times that of LiNbO_3 ($M_{\text{LN}}=29.2\times 10^{15}$ s^3kg^{-1}) [22]. GaP can also epitaxially be grown on silicon and is commercially available in 12" wafers (from NAsP GmbH) so is a scalable material platform.

On the acoustic material side, we choose piezoelectric ZnO as the material to generate phononic modes. The ZnO film can be readily sputtered on the GaP substrate, as described in the main text, with excellent piezoelectricity.

Therefore, the hybrid ZnO-GaP platform we demonstrate offers high performance, easiness of fabrication, and potential of scalability. Our results of the ZnO-GaP platform can inspire other hybrid material platforms that combine optimal material properties depending on different application scenarios.

4 (part I). What are the design tolerances?

Our response: The critical parameter to ensure phase-matching in our design is the matching between the acoustic mode wavenumber/frequency and the optical mode wavenumber/frequency, which are determined by the optomechanical waveguide geometry. Given a waveguide geometry, the phase-matching condition can be achieved by finding a pair of optical modes and an acoustic mode on the dispersion curves (Fig. 1b and 1c), and then design the IDT to excite the right acoustic frequency. From the design perspective, there is sufficient tolerance.

Below, we show the phase-matching condition for different waveguide widths. In this case, the design tolerance for the waveguide width is over 100 nm.

Waveguide width (μm)	Optical modes			Acoustic modes	
	β_1 ($1/\mu\text{m}$)	β_2 ($1/\mu\text{m}$)	$\Delta\beta$ ($1/\mu\text{m}$)	K ($1/\mu\text{m}$)	Ω (GHz)
1.00	7.25	10.45	3.20	3.20	2.60
1.05	7.50	10.50	3.00	3.00	2.45
1.10	7.71	10.52	2.81	2.81	2.37

The geometric parameters of the fabricated device, however, are inevitably deviated from the design because of the fabrication imprecision. Therefore, in the experiment, we need to search for the phase-matching condition by varying the acoustic frequency in a range. The range is limited by the IDT bandwidth, which is narrow for optimizing impedance matching but caused ideal phase-matching not to be achieved in our current devices. In future devices, we will design IDTs with wide bandwidth and add tuning mechanism (see below) to achieve phase matching.

4 (part II). Is there any tuning mechanism to meet the phase-matching conditions? It is noted that the phase mismatch is one of the main reasons for the limited conversion efficiency. So this is essential information.

Our response: Although the device is theoretically designed to achieve phase matching in a three-wave mixing configuration: two optical waves and one acoustic wave, the fabricated device deviates from the design. To achieve phase matching, we can detune the optical pumping and the acoustic frequency relative to the optical and acoustic resonance of the ring resonator. However, since the resonance frequencies of the ring are determined by its geometric parameters and are not tunable in our current devices, ideal phase matching is difficult to achieve, as we discussed in the manuscript.

Therefore, we agree with the reviewer that additional tuning mechanisms will be essential in future improved devices. Thermal tuning will be a convenient method to tune the optical and acoustic resonances by heating the material. To do so, one or multiple microheaters can be integrated in the proximity of the ring resonator. Besides, it is also possible to tune the optical resonance by using the electro-optic property of GaP, although it is relatively weak [1]. We will explore these options in future studies.

[1] D. F. Nelson, E. H. Turner; Electro-optic and Piezoelectric Coefficients and Refractive Index of Gallium Phosphide. *Journal of Applied Physics* 1 June 1968; 39 (7): 3337–3343.

5. What are the other reasons for the limited conversion efficiency? Can it improve at lower temperatures?

Our response: The conversion efficiency is defined as the power ratio between the converted mode (TE_2) and the input mode (TE_0):

$$\eta_i = \frac{P_{TE_2}}{P_{TE_0}}$$

where P_{TE_2} and P_{TE_0} are the generated TE_2 power and the input TE_0 power, respectively. Lowering the temperature does not have a direct impact on η_i , which depends on the phase-matching condition. What improves at a lower temperature is the electromechanical conversion efficiency of the IDT. In our current experiment, the electromechanical conversion efficiency is > 90% at a low temperature of 4 K when the IDT aluminum fingers become lower (not yet superconducting). It leads to better impedance matching and less resistive dissipation to improve the electromechanical conversion efficiency.

6. It would be better to add some parts of the supplementary information in the main text to explain (i) the fitted method, (ii) the interaction between the modes, and (iii) how the conversion efficiency is calculated.

Our response: We have edited the main text to include the following information, as suggested by the reviewer:

When the conversion efficiency is relatively low, the conversion to the TE_2 mode can be considered as an additional attenuation to the TE_0 mode pump, which we model with an acoustic power-dependent attenuation $b(P_a)$ in addition to the static intrinsic attenuation of the TE_0 mode. Incorporating $b(P_a)$ in the standard CMT, we reach the expressions for the TE_0 and TE_2 mode amplitudes, as detailed in Supplementary Note I, which have been used to fit the results in Fig. 4d and e. Our model captures these effects and fits the data with good agreement, allowing us to extract the value of $P_{\pi/2} = 1.6$ mW.

7. In supplementary information IV, it is noted that there is a reflection of the acoustic wave. How does this impact efficiency?

Our response: The reflected acoustic wave mentioned in the Supplementary Information is the acoustic wave that is reflected from the IDT at the other end (opposite to the end the acoustic wave is launched) of the bus waveguide. The reflection of the acoustic wave may couple to the OMR in the reverse direction to the desired (forward) acoustic wave. When both the forward and the reflected acoustic waves are present in the OMR, their interference may reduce the conversion efficiency.

8. What is phononic resistance?

Our response: We assume, by “phononic resistance”, the reviewer perhaps is referring to the mechanical “resistive elements” in the modified-BVD (mBVD) model. Here, we include the mBVD effective circuit from the Supplementary Information in the following.

FIG. S4. (a) The mBVD model that is used to fit the RF reflection spectrum (S_{11}). (b) The ratio of electrical power dissipated on the mechanical lumped element (R_m), which corresponds to the power converted to the mechanical power. (c) The measured S_{11} spectrum (solid blue line) and the mBVD model fitting (black dashed line). (d) The real and imaginary parts of the S_{11} spectrum. The dashed lines are the mBVD fitting.

The physical meaning of the resistive elements in the mBVD model, R_m , C_m , and L_m , as shown in Fig. S4(a), representing the overall impedance of the piezoelectric transducer. Our fitting to the measured S_{11} spectrum yields: $R_m = 50 \Omega$, $C_m = 0.0013 pF$, and $L_m = 3000 nH$. Therefore, the total impedance of the mechanical lumped elements, by design, is matched to the characteristic RF source impedance 50Ω , which results in a $> 90\%$ electromechanical conversion efficiency, as shown in Fig. S4(b).

9. What is the relationship between Q_{at} and Q_{oi} ?

Our response: There are four quality factors mentioned in the main text: acoustic loaded quality factor Q_{al} , acoustic intrinsic quality factor Q_{ai} , optical loaded quality factor Q_{ol} , and optical intrinsic quality factor Q_{oi} . The conversion between loaded and intrinsic quality factor is done by using the following formula:

$$Q_{ai(oi)} = \frac{2Q_{al(ol)}}{1 + \sqrt{T_0}}$$

Where T_0 is the transmission at the resonance frequency through the bus waveguide.

10. Finally, it is written, "The resonance signal is stronger at higher temperatures ... resulting in a better impedance matching". Is this effect taken into account during the design phase?

Our response: We think the reviewer is referring to the acoustic signal resonance, as in the main text:

*"The (acoustic) resonance signal is stronger at **lower** temperature..."*,

So it is stronger at lower temperatures rather than at higher temperatures.

Yes, this effect is already considered during the design phase. The small aperture IDT that is on the waveguide is, in fact, designed not impedance-matched at room temperature but only impedance-matched at cryogenic temperature to achieve optimal overall efficiency.

11. The paper doesn't cover noise and added noise. As measurements were actually performed, a detailed analysis would also help understand intrinsic efficiency.

Our response: We thank the reviewer for the suggestions. Following is the additional discussion about cooperativity and added noise we added in the revised Supplementary Information.

FIG. S3. General schematic for microwave-to-optical conversion.

A general schematic of the microwave-to-optical conversion of our device is shown in Fig. S3. Here, the microwave-to-optical conversion is a two-stage conversion, where the microwave signal is coupled to the mechanical resonator via electromechanical coupling with coupling rate g_{EM} , and the mechanical resonator is coupled to the optical resonator via optomechanical coupling with coupling rate g_{OM} . In this section, we will characterize key parameters related to the microwave-to-optical transduction. The following calculations are based on the work of Wu et al. [1], Han et al. [2] and Aspelmeyer et al. [3].

A. Calculating Cooperativities

A general form of the cooperativity is expressed as

$$C_{ij} = \frac{4g_{ij}^2}{k_i k_j}$$

where $i, j = (o, m, e)$ for two-stage microwave-to-optical transduction, namely, electromechanical (EM) and optomechanical (OM) coupling. And $k_i (k_j)$ is the mechanical damping rate of the $i^{th} (j^{th})$ stage. C_{EM} is the cooperativity between the microwave and the mechanical resonator, and the C_{OM} is the cooperativity between the optical and the mechanical resonator.

1. Calculating C_{EM}

First, we calculate the C_{EM} using the expression

$$C_{EM} = \frac{4g_{EM}^2}{\gamma_m k_e}$$

where g_{EM} is the electromechanical coupling rate, γ_m is the mechanical energy loss rate (unit: $\text{Hz} \cdot 2\pi$), and k_e is the electrical decay rate (FWHM). The g_{EM} (unit: $\text{Hz} \cdot 2\pi$) can be extracted from the modeled effective circuit in [3] using

$$g_{EM} = \frac{\sqrt{k_T^2} \omega_m}{2}$$

$$k_T^2 = \frac{C_m}{C_m + C_p + C_T}$$

where k_T^2 is the reduced piezoelectric coupling strength and $\omega_m = \omega_{LC}$ is the resonance frequency of the RF reflection spectrum. In our model, C_T is absorbed into C_p since the two capacitors are connected in parallel. The modified Butterworth Van Dyke (mBVD) model of the piezoelectric circuit for our system is shown in Fig. S4(a). The electrical decay rate k_e can be extracted from the FWHM of the RF reflection spectrum at resonance frequency, which is shown in Fig. S4(c).

We use the mBVD model to fit the S_{11} data to extract C_p and C_m . Fig. S4(c)(d) show the fitting result of the mBVD model. The fitted $C_p = 0.82 \text{ pF}$ and $C_m = 0.0013 \text{ pF}$. The fitting result is consistent with the measured data, and we can use the extracted capacitance value to calculate

$$k_T^2 = \frac{C_m}{C_m + C_p + 0} = \frac{0.0013 \text{ pF}}{0.82 \text{ pF} + 0.0013 \text{ pF}} \approx 1.582 \times 10^{-3}$$

$$\frac{\omega_m}{2\pi} = 2.556 \text{ (GHz)}$$

$$g_{EM} = \frac{\sqrt{k_T^2} \omega_m}{2} = 319.5 \left(\frac{\text{Mrad}}{\text{s}} \right)$$

$$k_e = 15 \text{ (MHz)}$$

$$\gamma_m = 1.1 \text{ (MHz)}$$

and we can calculate the C_{EM} as

$$C_{EM} = \frac{4g_{EM}^2}{\gamma_m \kappa_e} = \frac{4 \times (319.5 \times 10^6)^2}{15 \times 10^6 \times 1.1 \times 10^6 (2\pi)^2} = 626$$

2. Calculating C_{OM}

Optomechanical cooperativity can be calculated as:

$$C_{OM} = \frac{4g_{OM}^2}{\gamma_m \kappa_o}$$

where g_{OM} is the pump-enhanced coupling rate $g_{OM} = g_0 \sqrt{n_{phot}}$, and n_{phot} is the intracavity photon number. Single photon optomechanical coupling rate g_0 can be simulated using finite element analysis (FEA). Experimentally, g_0 can be extracted from the measured data and $\sqrt{n_{phot}}$ can be calculated from the optical pump power. γ_m can be extracted from the acoustic quality factor, and κ_o can be extracted from the optical quality factor. The phonon-flux normalized optomechanical coupling rate $g_{extract}/\sqrt{\hbar\Omega} = 230 \text{ (mm}^{-1}\sqrt{W^{-1}})$ and we can calculate the pump-enhanced coupling rate as

$$g_{OM} = g_0 \sqrt{N_{phot}} = \left(\frac{g_{extract}}{\sqrt{\hbar\Omega}} \sqrt{P_a \times \pi D} \right) \times \sqrt{N_{phot}}$$

where D is the diameter of the OM ring and P_a is the RF pump power and Ω is the frequency of the L_2 mode. We first perform the following calculations.

$$\begin{aligned} g_0 \sqrt{N_{phot}} &= \frac{g_{extract}}{\sqrt{\hbar\Omega}} \sqrt{1.5 \times 10^{-3} \times 200 \times 10^{-3} \pi \sqrt{N_{phot}}} \\ &= 5.7 \sqrt{N_{phot}} = 293 \times 10^6 = g_{OM} \\ \gamma_m &= 1.1 \text{ MHz} \\ \kappa_o &= 2.43 \text{ GHz} \end{aligned}$$

And we can calculate C_{OM} as

$$C_{OM} = \frac{4g_{OM}^2}{\gamma_m \kappa_e} = \frac{4 \times (293 \times 10^6)^2}{1.1 \times 10^6 \times 2.43 \times 10^9 (2\pi)^2} = 3.35$$

B. Added noise

Following the treatment in reference [1], the added noise N arises from two main contributions in our platform: optical noise contributed by the stokes scattering N_o and the thermomechanical noise from the pumped phonon N_m . The optical added noise can be expressed as

$$N_o = \frac{1}{\eta_e} \frac{C_{OM} \mathcal{L}^2}{C_{EM}}$$

where the electrical coupling efficiency is given by $\eta_e = \frac{Z_{tx}}{Z_{tx} + R_s}$. Z_{tx} is the characteristic impedance and R_s is the serial resistance of the mBVD model. The optical-cavity Lorentzian sideband amplitudes are expressed as

$$\mathcal{L}_{\pm}^2 = \frac{\left(\frac{\kappa_o}{2}\right)^2}{\left(\frac{\kappa_o}{2}\right)^2 + (\Delta \pm \omega_m)^2}$$

where $\Delta = \omega_{pump} - \omega_0$ is the laser detuning from cavity resonance, κ_o is the optical mode decay rate, ω_m is the acoustic resonance frequency. In our device, the calculated Stokes sideband amplitude $\mathcal{L}_{-}^2 = 0.18$, and we can use the information calculated so far to calculate optical added noise as

$$N_o = \frac{1}{0.72} \frac{(3.35 \times 0.18)}{626} = 0.0013$$

The mechanical thermal noise can be expressed as

$$N_m = \frac{1}{\eta_e} \frac{n_m}{C_{EM}} = \frac{1}{0.72} \frac{33}{626} = 0.07$$

where $n_m(\omega) = [e^{\hbar\omega/(k_B T)} - 1]^{-1}$ is given by the Bose-Einstein distribution and $\omega = \omega_m$. And the total added noise is the sum of the two contributions: $N_{add} = N_o + N_m = 0.0713$.

- [1] M. Wu, E. Zeuthen, K. C. Balram, and K. Srinivasan, "Microwave-to-optical transduction using a mechanical supermode for coupling piezoelectric and optomechanical resonators," *Phys. Rev. Appl.*, vol. 13, p. 014027, Jan 2020.
- [2] X. Han, W. Fu, C.-L. Zou, L. Jiang, and H. X. Tang, "Microwave-optical quantum frequency conversion," *Optica*, vol. 8, pp. 1050–1064, Aug 2021.
- [3] M. Aspelmeyer, T. J. Kippenberg, and F. Marquardt, "Cavity optomechanics," *Reviews of Modern Physics*, vol. 86, pp. 1391–1452, dec 2014.

12. From the abstract. "The conversion efficiency will reach unity when ideal phase-matching conditions are achieved." We instead know that the conversion efficiency depends on several parameters, such as microwave and acoustic losses, pump power, temperature, etc. Are all these parameters considered in the analysis? Will these parameters be considered for optimization?

Our response: Yes, we have considered these parameters in the analysis. The main limitation factor is the phase mismatch of the photonic and phononic mode inside the ring, we plan to address this issue with a phase tuning mechanism. The microwave and acoustic losses can be compensated with increased input power when it is considered as the pump.

Reviewer #2:

In the manuscript entitled "Optomechanical ring resonator for efficient microwave-optical frequency conversion," the authors demonstrate a new electro-optomechanical system based that utilizes suspended GaP based waveguides to confine both optical and acoustic waves. They use ZnO electromechanical transducers to launch acoustic waves into this system, which produce resonantly enhanced inter-modal Brillouin scattering within an optomechanical ring resonator. The work has several noteworthy novel aspects surrounding the use of new materials and their use of resonantly enhanced Brillouin scattering to mediate efficient scattering. However, there are several issues that should be addressed before I can support publication.

Our response: We thank the reviewer for the positive comment on the novel aspects of our work.

These include:

- In reading the manuscript, it was easy to miss the fact that the waveguide is suspended. Also, motivation for the use of a suspended waveguide was either subtle or not explained. I think this is a crucial part of their system and requires a bit more motivation and explanation in the main body of the text. Perhaps just one or two more sentences would be sufficient.

Our response: We thank the reviewer for the suggestion. We have added the following discussions in the main text.

The silicon under the GaP is undercut to suspend the whole OMR device in air to minimize the phononic loss through the substrate.

- Why is it necessary use ZnO as a piezoelectric transducer material? Is the piezo coefficient of GaP too small? It would be helpful to explain/motivate this decision further in the text.

Our response: The ZnO layer is necessary to generate acoustic waves efficiently. GaP indeed has piezoelectricity, but the non-vanishing coefficient ($e_{14} = -0.10 \text{ C/m}^2$) [1] is too small, more than ten times less than ZnO ($e_{33} = 1.32 \text{ C/m}^2$) [2]. Our experiment has shown that, without ZnO, the IDT on GaP has a very low electromechanical conversion efficiency which is barely measurable. To emphasize the necessity of using ZnO, we added the following discussion in our main text.

ZnO's strong piezoelectricity is used to efficiently generate the acoustic waves in our experiment, although GaP has a non-zero piezoelectric coupling coefficient, it is too weak for us to measure RF resonance and transmission using only GaP.

[1] D. F. Nelson, E. H. Turner; Electro-optic and Piezoelectric Coefficients and Refractive Index of Gallium Phosphide. *Journal of Applied Physics* 1 June 1968; 39 (7): 3337–3343.

[2] Fraga, M.A., Furlan, H., Pessoa, R.S. *et al.* Wide bandgap semiconductor thin films for piezoelectric and piezoresistive MEMS sensors applied at high temperatures: an overview. *Microsyst Technol* **20**, 9–21 (2014)

- What is the limitation of phonon losses in the current experiments? Why do you think your Q-factor didn't increase much at low temperatures? Some elaboration on this point in the main text would be helpful.

Our response: While the thermomechanical phonon loss will drop precipitately with lowering temperature, the material's intrinsic loss and device structure loss will not change with temperature. We think the latter are the dominant phonon loss mechanisms in the GaP devices, so the Q-factor won't increase much at low temperature. Our recent work [1] on boron-doped GaP material characterization has shown that there is considerable Si clustering at the GaP-Si interface, which can be a major contribution to phonon loss even after suspending the GaP layer.

We have included the following explanation in the main text.

We have identified material defects at the interface between silicon and GaP in our material system⁶¹. We suspect these defects are the dominant phononic loss channel that is temperature independent. As a result, the improvement in the phononic Q-factor at cryogenic temperatures is not significant.

[1] N. S. Yama, I.-T. Chen, et al, "Silicon-lattice-matched boron-doped gallium phosphide: A scalable acousto-optic platform," arXiv: 2305.11436 (2023)

• I'm perplexed by the examination of the time reversal (TR) symmetry of the system. All classical and quantum systems exhibit time-reversal symmetry. There is no controversy here, as time reversal of any Hamiltonian yields the same behavior unless we fail to reverse time for one of the field quantities (e.g. magnetic field). Isn't it really nonreciprocity that you examine?

Our response: We agree with the reviewer that, as a whole, our system is completely reciprocal because it involves no magnetic field and no nonlinearity. However, there is a strong technical interest in achieving optical non-reciprocity for optical isolation and directional circulation. Therefore, from the optics' perspective only, the traveling acoustic wave, with a fixed propagation direction, breaks the time-reversal symmetry for light propagation. Our device, with symmetric optical and acoustic ports, provides a versatile platform to examine the reciprocity (or optical non-reciprocity).

To reiterate:

- I think we can all agree that TR symmetry is always holds if we can apply time reversal to all dynamical variables.
- B-fields do not break time-reversal symmetry; only if we do time reversal incorrectly does one come to the incorrect conclusion.

Is there some fundamental physics you seek to discover here? Please provide further justification. Otherwise, I would recommend removing the discussion of time reversal symmetry or pushing to supplement.

Our response: As discussed above, there is an application interest to induce *optical* non-reciprocity without using magnetic field or materials for optical isolation and directional circulation. Please see references [1-6] below. In our system, the traveling acoustic wave generates a dynamic modulation in the medium and induces optical non-reciprocity when only considering the propagation of the optical wave. Of course, if the acoustic wave propagation is also reversed during full time-reversal, the device should be reciprocal, as we have demonstrated. We think our results in Fig. 3 are valuable in that our device uniquely has a symmetrical arrangement of input/output optical and acoustic ports, allowing for reversal of both optical and acoustic wave propagation, which has not been realized in any previous devices.

[1] Yu, Z., Fan, S. Complete optical isolation created by indirect interband photonic transitions. *Nature Photon* **3**, 91–94 (2009).

[2] Lira, H., Yu, Z., Fan, S. & Lipson, M. Electrically Driven Nonreciprocity Induced by Interband Photonic Transition on a Silicon Chip. *Phys. Rev. Lett.* **109**, (2012).

[3] Sohn, D. B., Kim, S. & Bahl, G. Time-reversal symmetry breaking with acoustic pumping of nanophotonic circuits. *Nat Photonics* 91–97 (2018).

[4] Sohn, D. B., Örsel, O. E. & Bahl, G. Electrically driven optical isolation through phonon-mediated photonic Autler–Townes splitting. *Nat Photonics* **15**, 822–827 (2021).

[5] Kittlaus, E. A. *et al.* Electrically driven acousto-optics and broadband non-reciprocity in silicon photonics. *Nat Photonics* **15**, 43–52 (2021).

[6] Tian, H. *et al.* Magnetic-free silicon nitride integrated optical isolator. *Nat Photonics* **15**, 828–836 (2021).

• To eliminate the ambiguity of their findings, I ask the authors please indicate the demonstrated optical and electromechanical cooperativities of this system.

Our response: Following the reviewer’s suggestion, we have included a detailed discussion on cooperativities in the Supplementary Information. Please see below:

FIG. S3. General schematic for microwave-to-optical conversion.

A general schematic of the microwave-to-optical conversion of our device is shown in Fig. S3. Here, the microwave-to-optical conversion is a two-stage conversion, where the microwave signal is coupled to the mechanical resonator via electromechanical coupling with coupling rate g_{EM} , and the mechanical resonator is coupled to the optical resonator via optomechanical coupling with coupling rate g_{OM} . In this section, we will characterize key parameters related to the microwave-to-optical transduction. The following calculations are based on the work of Wu et al. [1], Han et al. [2] and Aspelmeyer et al. [3].

A. Calculating Cooperativities

A general form of the cooperativity is expressed as

$$C_{ij} = \frac{4g_{ij}^2}{k_i k_j}$$

where $i, j = (o, m, e)$ for two-stage microwave-to-optical transduction, namely, electromechanical (EM) and optomechanical (OM) coupling. And k_i (k_j) is the mechanical damping rate of the i^{th} (j^{th}) stage. C_{EM} is

the cooperativity between the microwave and the mechanical resonator, and the C_{OM} is the cooperativity between the optical and the mechanical resonator.

1. Calculating C_{EM}

First, we calculate the C_{EM} using the expression

$$C_{EM} = \frac{4g_{EM}^2}{\gamma_m k_e}$$

where g_{EM} is the electromechanical coupling rate, γ_m is the mechanical energy loss rate (unit: $\text{Hz} \cdot 2\pi$), and k_e is the electrical decay rate (FWHM). The g_{EM} (unit: $\text{Hz} \cdot 2\pi$) can be extracted from the modeled effective circuit in [3] using

$$g_{EM} = \frac{\sqrt{k_T^2} \omega_m}{2}$$

$$k_T^2 = \frac{C_m}{C_m + C_p + C_T}$$

where k_T^2 is the reduced piezoelectric coupling strength and $\omega_m = \omega_{LC}$ is the resonance frequency of the RF reflection spectrum. In our model, C_T is absorbed into C_p since the two capacitors are connected in parallel. The modified Butterworth Van Dyke (mBVD) model of the piezoelectric circuit for our system is shown in Fig. S4(a). The electrical decay rate k_e can be extracted from the FWHM of the RF reflection spectrum at resonance frequency, which is shown in Fig. S4(c).

We use the mBVD model to fit the S_{11} data to extract C_p and C_m . Fig. S4(c)(d) show the fitting result of the mBVD model. The fitted $C_p = 0.82 \text{ pF}$ and $C_m = 0.0013 \text{ pF}$. The fitting result is consistent with the measured data, and we can use the extracted capacitance value to calculate

$$k_T^2 = \frac{C_m}{C_m + C_p + 0} = \frac{0.0013 \text{ pF}}{0.82 \text{ pF} + 0.0013 \text{ pF}} \approx 1.582 \times 10^{-3}$$

$$\frac{\omega_m}{2\pi} = 2.556 \text{ (GHz)}$$

$$g_{EM} = \frac{\sqrt{k_T^2} \omega_m}{2} = 319.5 \left(\frac{\text{Mrad}}{\text{s}} \right)$$

$$k_e = 15 \text{ (MHz)}$$

$$\gamma_m = 1.1 \text{ (MHz)}$$

and we can calculate the C_{EM} as

$$C_{EM} = \frac{4g_{EM}^2}{\gamma_m k_e} = \frac{4 \times (319.5 \times 10^6)^2}{15 \times 10^6 \times 1.1 \times 10^6 (2\pi)^2} = 626$$

2. Calculating C_{OM}

Optomechanical cooperativity can be calculated as:

$$C_{OM} = \frac{4g_{OM}^2}{\gamma_m \kappa_o}$$

where g_{OM} is the pump-enhanced coupling rate $g_{OM} = g_0 \sqrt{n_{phot}}$, and n_{phot} is the intracavity photon number. Single photon optomechanical coupling rate g_0 can be simulated using finite element analysis (FEA). Experimentally, g_0 can be extracted from the measured data and $\sqrt{n_{phot}}$ can be calculated from the optical pump power. γ_m can be extracted from the acoustic quality factor, and κ_o can be extracted from the optical quality factor. The phonon-flux normalized optomechanical coupling rate $g_{extract}/\sqrt{\hbar\Omega} = 230 \text{ (mm}^{-1}\sqrt{W}^{-1}\text{)}$ and we can calculate the pump-enhanced coupling rate as

$$g_{OM} = g_0 \sqrt{N_{phot}} = \left(\frac{g_{extract}}{\sqrt{\hbar\Omega}} \sqrt{P_a} \times \pi D \right) \times \sqrt{N_{phot}}$$

where D is the diameter of the OM ring and P_a is the RF pump power and Ω is the frequency of the L_2 mode. We first perform the following calculations.

$$\begin{aligned} g_0 \sqrt{N_{phot}} &= \frac{g_{extract}}{\sqrt{\hbar\Omega}} \sqrt{1.5 \times 10^{-3} \times 200 \times 10^{-3} \pi \sqrt{N_{phot}}} \\ &= 5.7 \sqrt{N_{phot}} = 293 \times 10^6 = g_{OM} \\ \gamma_m &= 1.1 \text{ MHz} \\ \kappa_o &= 2.43 \text{ GHz} \end{aligned}$$

And we can calculate C_{OM} as

$$C_{OM} = \frac{4g_{OM}^2}{\gamma_m \kappa_o} = \frac{4 \times (293 \times 10^6)^2}{1.1 \times 10^6 \times 2.43 \times 10^9 (2\pi)^2} = 3.35$$

B. Added noise

Following the treatment in reference [1], the added noise N arises from two main contributions in our platform: optical noise contributed by the stokes scattering N_o and the thermomechanical noise from the pumped phonon N_m . The optical added noise can be expressed as

$$N_o = \frac{1}{\eta_e} \frac{C_{OM} \mathcal{L}_{\pm}^2}{C_{EM}}$$

where the electrical coupling efficiency is given by $\eta_e = \frac{Z_{tx}}{Z_{tx} + R_s}$. Z_{tx} is the characteristic impedance and R_s is the serial resistance of the mBVD model. The optical-cavity Lorentzian sideband amplitudes are expressed as

$$\mathcal{L}_{\pm}^2 = \frac{\left(\frac{\kappa_o}{2}\right)^2}{\left(\frac{\kappa_o}{2}\right)^2 + (\Delta \pm \omega_m)^2}$$

where $\Delta = \omega_{pump} - \omega_0$ is the laser detuning from cavity resonance, κ_o is the optical mode decay rate, ω_m is the acoustic resonance frequency. In our device, the calculated Stokes sideband amplitude $\mathcal{L}_{\pm}^2 = 0.18$, and we can use the information calculated so far to calculate optical added noise as

$$N_o = \frac{1}{0.72} \frac{(3.35 \times 0.18)}{626} = 0.0013$$

The mechanical thermal noise can be expressed as

$$N_m = \frac{1}{\eta_e} \frac{n_m}{C_{EM}} = \frac{1}{0.72} \frac{33}{626} = 0.07$$

where $n_m(\omega) = [e^{\hbar\omega/(k_B T)} - 1]^{-1}$ is given by the Bose-Einstein distribution and $\omega = \omega_m$. And the total added noise is the sum of the two contributions: $N_{add} = N_o + N_m = 0.0713$.

[1] M. Wu, E. Zeuthen, K. C. Balram, and K. Srinivasan, "Microwave-to-optical transduction using a mechanical supermode for coupling piezoelectric and optomechanical resonators," *Phys. Rev. Appl.*, vol. 13, p. 014027, Jan 2020.

[2] X. Han, W. Fu, C.-L. Zou, L. Jiang, and H. X. Tang, "Microwave-optical quantum frequency conversion," *Optica*, vol. 8, pp. 1050–1064, Aug 2021.

[3] M. Aspelmeyer, T. J. Kippenberg, and F. Marquardt, "Cavity optomechanics," *Reviews of Modern Physics*, vol. 86, pp. 1391–1452, dec 2014.

• The authors indicate that they use a "Flux normalized coupling coefficient," but then show a power normalized coupling on page 8. Is this a mistake?

Our response: There is indeed a mistake in the main text. It should be:

From the fitting, we extract the flux-normalized coupling coefficient $\frac{g}{\sqrt{\hbar\Omega}} = 230 \text{ mm}^{-1}\sqrt{\text{W}^{-1}}$.

The flux-normalized coupling coefficient g is normalized to both optical power and phonon number flux. The definition is given in Supplementary Information section I. Because g is normalized to the square root of phonon number flux, it has a unit of $\text{mm}^{-1}\text{W}^{-1/2}$.

• A 90% IDT to phonon conversion efficiency is quoted. The definitions of conversion efficiency are very important and there is some ambiguity. What is the IDT conversion efficiency? What is the IDT power? Would the quoted 90% IDT to phonon conversion efficiency equate to a 90% percent quantum efficiency at zero Kelvin?

Our response: The IDT conversion efficiency is defined as the power ratio of the power on mechanical lumped elements P_m and the input power P_{in} , in the modified-Butterworth Ven Dyke model (mBVD), as shown in Fig. S4(a) and below. The effective admittance of the mechanical lumped elements is given as $Y_m = (R_m + j\omega_m L_m + 1/j\omega_m C_m)^{-1}$. The input power that is transferred to Y_m can be expressed as

$$P_m = V_{Y_m}^2 \cdot Y_m$$

where V_{Y_m} is the voltage across effective admittance Y_m . In addition to Y_m , the mBVD circuit also includes the serial resistor R_s , serial inductor L_s , parallel resistor R_p , parallel capacity C_p . The mBVD circuit can be simplified as a serial impedance $Z_s = R_s + j\omega L_s$ connected to an effective load impedance Z_{load} in series. The total impedance of the mBVD circuit can be expressed as

$$Z_{total} = Z_s + Z_{load},$$

where $Z_{load} = \left(\frac{1}{Z_p} + Y_m\right)^{-1}$ and $Z_p = \frac{1}{j\omega C_p + 1/R_p}$. We can calculate the voltage across Z_{load} using the expression

$$V_{load} = V_{in} - V_s$$

where V_{in} and V_s are the input voltage and the voltage across Z_s . Since Y_m is parallelly connected to R_p and C_p , the voltage across mechanical elements $V_{Y_m} = V_{load}$. Finally, we use P_m/P_{in} as the IDT conversion efficiency.

Fitting the measured S_{11} spectrum yields: $R_m = 50 \Omega$, $C_m = 0.0013 \text{ pF}$, and $L_m = 3000 \text{ nH}$, the fitting result is shown as the dashed line in Fig. S4(c)(d), which agrees with the measured data. The physical meaning of the resistive elements in the mBVD model, R_m , C_m , and L_m , as shown in Fig. S4(a), represent the overall impedance of the piezoelectric material. In our fitting, 90% of the input RF power is translated to Y_m , as shown in Fig. S4(b). The absolute RF input power here is -7 dBm. The 90% IDT conversion efficiency means that the 90% of the RF input power is being converted to mechanical lumped elements.

FIG. S4. (a) The mBVD model that is used to fit the RF reflection spectrum (S_{11}). (b) The ratio of electrical power dissipated on the mechanical lumped element (R_m), which corresponds to the power converted to the mechanical power. (c) The measured S_{11} spectrum (solid blue line) and the mBVD model fitting (black dashed line). (d) The real and imaginary parts of the S_{11} spectrum. The dashed lines are the mBVD fitting.

• I object to the manner in which the authors appear to claim a P_π of 0.1 mW since they have not achieved it. They appear to claim a level of performance in conjunction with a speculative qualifier "if phase matching is achieved". This is highly problematic and they must adjust their claims to reflect what they have achieved. Perhaps a speculative statement regarding future performance could be included in the

conclusions. However, the authors must make it clear that they have not accomplished this result. It would be more appropriate to put the P_{π} that they have actually achieved. It is currently absent.

Our response: We agree with the reviewer's comment. $P_{\pi/2} = 0.1 \text{ mW}$ is calculated by assuming the ideal phase matching is achieved. To avoid misunderstanding, we have revised the discussion as the following:

We can calculate the critical acoustic power needed to achieve unity conversion efficiency in the OMR if phase matching is achieved:

$$P_{\pi/2} = \left(\frac{\sqrt{\hbar\Omega}}{2gD} \right)^2 = 0.1 \text{ mW}$$

where $D = 200 \mu\text{m}$ is the diameter of the OMR.

• Additionally, who uses this metric? Feels like it comes out of thin air to bolster claims. Also, what physical attributes lower P-pi in this system?

Our response: The $P_{\pi/2}$ metric is frequently used in the context of acousto-optic mode conversion, as in the reference [1-4] below. The lower $P_{\pi/2}$ in our system is attributed to the strong optomechanical coupling coefficient from the co-resonance and co-confinement of the photonic and phononic modes.

[1] Sarabalis, C. J. *et al.* Acousto-optic modulation of a wavelength-scale waveguide. *Optica* **8**, (2021).

[2] C. Duchet, C. Brot, and M. Di Maggio, "Interdigital transducer for acousto-optic tunable filter on LiNbO₃," *Electron. Lett.* **31**, 1235–1237 (1995).

[3] I. Hinkov, V. Hinkov, and E. Wagner, "Low power integrated acousto-optical tunable filters in first telecommunication window," *Electron. Lett.* **30**, 1884–1885 (1994).

[4] Y. Ohmachi and J. Noda, "LiNbO₃ TE-TM mode converter using collinear acousto-optic interaction," in *IEEE Journal of Quantum Electronics*, vol. 13, no. 2, pp. 43-46, February 1977, doi: 10.1109/JQE.1977.1069276.

• One of the key claims of this paper needs to be amended. In particular, the following statement is not correct: "In conclusion, we have demonstrated the first OMR in which photonic and phononic modes are co-resonantly coupled to achieve efficient optomechanical mode conversion with a record low critical acoustic power ($P_{\pi/2}$)." This paper by Yoon [Yoon, et al. *Optica* 10.1 (2023): 110-117] have previously achieved this condition through an intermodal Brillouin scattering process, and should be cited. Perhaps it would be appropriate to indicate that this is the first micro-scale system. On the other hand, the authors didn't really achieve the quoted record Ppi value or the resonance condition. So I think that this claim is not really appropriate.

Our response: We thank the reviewer for pointing out the prior reference and we have cited it in the revised main text. We have revised the conclusion with a reduced claim as follows:

In conclusion, we have demonstrated the first microscale OMR in which photonic and phononic modes are co-resonantly coupled to achieve efficient optomechanical mode conversion.

In conclusion, I believe that this system is novel and interesting, however, I would need to see how the authors address the above comments and revise the claims in their manuscript before I could support

publication.

Our response: We thank the reviewer for the comment on our work. We hope our responses are satisfactory to the reviewer.

Reviewer #3:

In ‘Optomechanical ring resonator for efficient microwave-optical frequency conversion’ the authors present a realisation of an efficient optomechanical conversion device, where travelling photons and phonons coexist in a ring resonator. As a result of the photoelastic and moving boundary coupling mechanisms, the confined phonons result in an inter-modal optical coupling. Through a series of optical and microwave-frequency tests the propagation-dependent phase-matching is analysed in the paper and the authors provide an outlook as to how the efficiency can be increased.

The manuscript presents an interesting solution to the challenge of efficiently converting microwave and optical photons, and provides results interesting to researchers developing integrated optomechanical and electromechanical transduction devices. The manuscript is, on the whole, clearly written and the explanation of the coupling mechanism and the required phase-matching is outlined fully.

The use of both propagating mechanical and optical states on the chip is an important step towards scalable integrated devices capable of quantum frequency conversion, and the device the authors present is a promising realization of this. Nonetheless, in order for me to recommend publication, a number of issues with the content of the manuscript, in particular on how the data is treated and presented and how the device is modelled need to be addressed first. I have outlined these issues and questions below.

Our response: We thank the reviewer for summary, the positive comment, and stating the importance of our work.

1, In the abstract the authors state ‘the optomechanical conversion between photonic modes to achieve a conversion efficiency of 8.2% at low acoustic pump power 1.6 mW.’ They need to clarify this statement. Are they referring to the intrinsic device efficiency here? What would the total efficiency of the device be?

Our response: The conversion efficiency in the abstract refers to the intrinsic conversion efficiency, which is defined in the Supplementary Information as below:

(page.3 in the Supplementary Information II)

The internal optomechanical conversion efficiency is defined as

$$\eta_i = \frac{P_{TE2}}{P_{TE0}}$$

where P_{TE2} is the optomechanical converted TE_2 mode power inside OMR, and P_{TE0} is the power of TE_0 mode that coupled into OMR. P_{TE0} is known because the laser power, grating coupler coupling efficiency, and the transmission of the TE_0 through port is known. So, to calculate the optomechanical conversion efficiency, we need to obtain the P_{TE2} .

To clarify, we revised the abstract as

to achieve **an internal** conversion efficiency of ...

The total device efficiency is defined as: $\eta_{tot} = \frac{P_{out}}{P_{laser}}$, where P_{out} is the device output optical power receiving from the grating coupler at TE_2 output port and P_{laser} is the input laser power. These are the optical powers after and before, respectively, the grating couplers, which each have an efficiency of -15 dB and can be improved.

The values of the power levels can be found in the Supplementary Information. Using this definition, we calculate $\eta_{tot} = \frac{6.84nW}{12mW} = 0.57 \times 10^{-6}$, when an acoustic pump power of 1.6 mW is used. We added the following calculation at the end of the bookkeeping section in the Supplementary Information.

The total conversion efficiency is defined as

$$\eta_{tot} = \frac{P_{out}}{P_{laser}}$$

where P_{out} is the output power of the device and P_{laser} is the input laser power. The calculated $\eta_{tot} = 0.57 \times 10^{-6}$ when an acoustic pump power of 1.6 mW is used.

2, The authors state additionally in the abstract that ‘The conversion efficiency will reach unity when ideal phase-matching is achieved.’ This statement is quite vague. Again, I assume they are discussing the internal conversion efficiency inside the ring resonator. However, even in this case I find this to be a potentially overblown statement. Do they expect with the loss rates in the system that every photon in the TE0 mode will be converted to a photon in the TE2 mode? In particular, would the large coupling rate of the TE0 mode to the resonant TE2 mode be extinguished in this case?

Our response: First, we agree that, when there is loss, the conversion efficiency will not reach unity. Therefore, we revise this statement to “Further improvement in the conversion efficiency will be achieved when ideal phase-matching is satisfied and when the system’s losses are low.”

Second, indeed, when the conversion efficiency is very high, the pump (TE0 mode) will be depleted, and the coupling coefficient will gradually reduce to zero. The overall (integrated) conversion efficiency will remain high, which is actually the desired situation for efficient mode conversion.

3, The authors do not give any information as to the value of kappa_a (the coupling from the phonon waveguide to the OMR in the device).

Our response: We thank the reviewer for pointing this out. The acoustic waveguide-OMR coupling coefficient κ_a is extracted by fitting the RF transmission spectrum of the OMR (Fig. 2d) and we obtain a value of $\kappa_a=0.8$. We have added this parameter to the parameter table.

4, In the main body of the text the intrinsic Q-factor of the optical mode is listed as 6.4×10^4 , however in the figure the mode is labelled with $Q_{ol} = 4.1 \times 10^5$. Firstly, is Q_{ol} the loaded quality factor? This is not explained. If this is indeed the case - it would be unphysical to have a loaded quality factor larger than the quality factor owing to intrinsic loss. This needs to be corrected.

Our response: We thank the reviewer for pointing this out. Indeed, Q_{ol} is the loaded quality factor. There is indeed an error in the label in Fig. 2c: Q_{ol} should be 4.1×10^4 , not 4.1×10^5 . After improving our measurement (see the response to the next comment), we now measure Q_{ol} to be 7.5×10^4 . We have revised the figure description to clarify.

Fig. 2 c. The loaded optical quality factor is $Q_{ol} = 7.5 \times 10^4$. d. The loaded acoustic quality factor is $Q_{al} = 2.3 \times 10^3$.

5, The optical transmission signal in figure 2c oscillates a lot, with the same periodicity in wavelength as for the TE₀ mode. The authors should outline the source of this oscillation.

Our response: The oscillation with a large free-spectral range (FSR) is induced by the Fabry-Perot effect due to the reflection between the fiber facets and the grating couplers. We have improved the measurement setup to remove it by moving the fiber much closer to the sample and remeasured the optical transmission. The new data is shown in the figure below. The loaded optical quality factor is improved to $Q_{ol} = 7.5 \times 10^4$.

6, In the book-keeping part of section II of the supplementary information, if i take $P_{\text{device}} = 0.03\text{uW}$ and normalise it by the grating coupler efficiency (0.03) and the outcoupling efficiency of the TE₂ mode (0.03) I get $P_{\text{TE2}} = 33 \text{ uW}$, rather than 22.8. There seems to potentially be a mistake here. Also the value of 3% for η_{OC} is very inconsistent with the value in table S2, where it is listed as $> 60\%$. The reasons for this mismatch should be explained. Finally - how is the outcoupling efficiency of the TE₀₂ mode measured?

Our response: We thank the reviewer for finding this error in our manuscript. The notation used in the Supplementary Information of the out-coupling efficiency is mislabeled. The correct labeling should be $\eta_{02} = 3.0\%$. The revised Supplementary Table S3 is as the following.

OMIC device parameters				
	Parameters	Unit	TE ₀	TE ₂
Multi-mode OMR	Wavenumber modes(β_i)	μm^{-1}	10.43	7.30
	Wavenumber difference ($\beta_0 - \beta_2$)	μm^{-1}		3.13
	Effective mode index(n_{eff})		2.558	1.784
	Group index(n_g)		3.245	3.315
	Hybrid TE ₀ -TE ₀ * (Δn_{eff})			0.013
	Multi and single mode waveguide width(w)	μm	1.01	0.50
	Ring diameter(D)	μm		200
	Ring total length(l_{tot})	μm		628
	Optical coupling length(L_{ii})	μm	$L_{00} = 50$	$L_{02} = 60$
	Optical coupling gap width(g_{ii})	μm	$g_{00} = 0.07$	$g_{02} = 0.20$
	Waveguide to OMR coupling efficiency(η_{ii})	%	$\eta_{00} = 90.0$	$\eta_{02} = 3.0$
	Single grating coupler efficiency(η_{GC})	%		3.0
	Optical FSR	nm	1.14	1.08
2nd-order Lamb mode (L_2)		Parameters	Unit	L_2
		Acoustic wave central frequency($\Omega/2\pi$)	GHz	2.56
		Acoustic wavenumber(κ)	μm^{-1}	3.13
		Acoustic waveguide width(w_a)	μm	1.01
		Acoustic waveguide tapered length(l_a)	μm	100
		Acoustic Coupling length(l_a)	μm	$l_a = \frac{2\pi}{\Delta K} = \frac{2\pi}{0.4} = 13$
		Acoustic Coupling gap width(g_a)	μm	0.2
		Simulated acoustic group velocity (V_g)	m/s	3424
		Measured acoustic group velocity (V_g)	m/s	3450
		Calculated Acoustic FSR	MHz	5.4
		Measured Acoustic FSR	MHz	5.5
		IDT pitch(Λ)	μm	2.0
	IDT aperture(W)	μm	15.0	
	IDT efficiency(η_a)	%	90.0	

Table S3. The phononic and photonic design parameters of the OMR.

For the TE_2 mode out-coupling efficiency, η_{02} , the device design does not allow us to measure it directly. Therefore, we use FDTD simulation to estimate it to be $\eta_{02} \sim 3.0\%$ as shown below.

Figure R1. FDTD simulation of the TE_2 mode out-coupler. From the simulation, we can see that at the coupling region, TE_2 mode can couple $\sim 10\%$ of $|E|^2$ to the output waveguide, but only $\sim 3\%$ of $|E|^2$ is

transmitted (indicated by the white arrow on the right) to the grating coupler due to the bending loss of the waveguide. From this simulation, we can estimate that at the output waveguide before the grating coupler, the maximum transmission efficiency $\eta_{02} = 3.0\%$.

As for the calculation of the optomechanical mode conversion efficiencies, we address these comments together in the comment below.

7, Similarly, if i follow the values for $\eta_{GC} = 0.03$ and $\eta'_{OC} = 0.9$ and $P_{laser} = 12e-3W$ - i get $P_{TE0} = 324 \text{ uW}$. With $P_{TE2} = 22.8 \text{ uW}$ - i get an efficiency of $22.8/324 = 0.07$, rather than 0.082. It should be noted that the coupling efficiencies are only quoted to one significant figure here, which causes a great degree of uncertainty in the device efficiency.

Our response: We thank the reviewer for spotting the discrepancy. We have corrected the calculation of the optomechanical mode conversion efficiency and updated it in the Supplementary Information. We also added a power flow diagram to help readers to track the calculation of the conversion efficiency. Below, we include the revise Supplementary Information for reference.

FIG. S2. TE_0 -to- TE_2 optomechanical conversion efficiency calculation flowchart. The red arrows represent the optical signal, and the blue arrow represents the electrical signal. The input laser power P_{laser} is coupled to the OMR through a grating coupler (GC) with a coupling efficiency η_{GC} . The waveguide-to-OMR coupling efficiency is η_{00} . The input TE_0 mode is then converted to TE_2 mode in the OMR with an intrinsic efficiency η_i . The TE_2 mode is then coupled out of the OMR with efficiency η_{02} and out of the chip from the grating coupler with efficiency η_{GC} . The output TE_2 optical power is labeled as P_{out} . P_{out} is then amplified by EDFA with a gain G_{EDFA} and beats with the AOFS in a directional coupler (DC). The beating optical signal P_{beat} is finally detected by the high-speed photoreceiver (HPR) with a conversion gain G_{HPR} and converted to electric signal P_{elec} and read out by the RSA.

The pre-factors 0.9 and 0.1 are the table-top directional coupler's transmission coefficient that is used to combine the AOFS and the device output signal.

$$P_{out} = \frac{P_{elec} R_{char} / G_{HPR}^2}{G_{EDFA} \times 0.9 \times 0.1 \times P_{AOFS}}$$

where $P_{elec} = -55$ dBm, the resulted $P_{out} = 6.16$ nW.

Now we can back-calculate the power inside the OMR by considering grating coupler efficiency η_{GC} and the TE_2 out-couple efficiency η_{02} using

$$P_{TE_2} = \frac{P_{out}}{\eta_{GC}\eta_{02}}$$

where $\eta_{GC} = 3.0\%$ and $\eta_{02} = 3.0\%$, respectively. The resulted optomechanically converted TE_2 power in the OMR $P_{TE_2} = 6.84\mu W$. On the input side, the TE_0 power in the ring can be calculate as

$$P_{TE_0} = P_{laser}\eta_{GC} \cdot \eta_{00}$$

where $P_{laser} = 12$ mW, and $\eta_{00} = 90\%$. The resulted input $P_{TE_0} = 324$ mW, and the intrinsic optomechanical conversion efficiency $\eta_i = 2.1 \pm 0.1$ %. The total conversion efficiency is defined as

$$\eta_{tot} = \frac{P_{out}}{P_{laser}}$$

where P_{out} is the output power from the device and P_{laser} is the lase input power. The calculated $\eta_{tot} = 0.57 \times 10^{-6}$ when the acoustic power

8, In figures 4 b and c what is the acoustic power used for these measurements?

Our response: The RF power used to excite the acoustic wave in Fig 4b and Fig 4c is -7.0 dBm, which corresponds to -7.5 dBm of acoustic power in the IDT, considering an electromechanical conversion efficiency of 90%.

9, In figure 4c, d and e are the powers in the y-axis the electrical powers or rather the optical powers from the device? This should be clarified. The plots appear inconsistent with a device efficiency of 0.08, as at the peak of efficiency they measure 400x more power in the P02(delta) mode than the P_02(Omega - delta) mode.

Our response: In Figure 4c, d, the power in the y-axis is the electric power detected by the high-speed photoreceiver (HPR) and recorded on the RSA. The $P_{02}(\delta)$ is the beating signal between the AOFS signal and the static TE_2 mode, which is converted to the actual input power (P_{TE_0}) of the TE_0 mode by considering the HPR conversion gain, the gain of the EDFA, and the AOFS signal power.

So the conversion efficiency, that is, the ratio between P_{TE_0} and P_{TE_2} , cannot be directly calculated by taking the ratio of $P_{02}(\delta)$ and $P_{02}(\Omega - \delta)$. The photoreceiver gain, EDFA gain, and reference signal power need to be considered to calculate the conversion efficiency. To clarify, we revise the main text as follows:

The internal conversion efficiency $\eta_i = P_{TE0}/P_{TE2}$ is calculated by converting $P_{02}(\delta)$ and $P_{02}(\Omega - \delta)$ to P_{TE0} and P_{TE2} after considering the photoreceiver gain, EDFA gain, and reference signal power (see Supplementary Information II).

10, With regards to the microwave-frequency S_{21} measurement - the authors state that the electromechanical conversion efficiency is 90% in the device, and yet they need to filter the VNA measurement heavily in order to see transmitted mechanical mode. These two facts seem inconsistent. Presumably if 90% of the microwave power was being converted into travelling phonons in the waveguide, one would expect all other parasitic contributions to be much smaller. Additionally, filtering in the time domain can leave artefacts on the spectral response, and as such the authors should include the unfiltered transmission spectrum.

Our response: The electromechanical conversion efficiency refers to the efficiency at the acoustic transducer, which includes the IDT and the ZnO on the wide acoustic waveguide. However, when the acoustic mode propagates to the narrow waveguide, it suffers a high loss due to mode conversion. Similarly, on the receiving end of the acoustic waveguide, the mode expander from the narrow waveguide to the wider receiver, which also includes the IDT and the ZnO, also induces loss. As a result, the overall signal of the detected L_2 mode is weak, requiring the time-gating filtering method to extract from a background of parasitic coupling. The filtering method we employ is simply to perform a Fourier transform of the S_{21} signal after proper time gating, which is a common practice in detecting acoustic ring resonance signals, as employed by other groups [1][2]. Below, we show the unfiltered and filtered transmission spectrum, which is also added to the revised Supplementary Information.

FIG. S5. Time-gating signal processing of the acoustic transmission $|S_{21}|$. (a) The unprocessed $|S_{21}|$ spectrum. (b) The $|S_{21}|$ in the time domain after Fourier transform. (c) The $|S_{21}|$ spectrum after time-gating

from $190 < t < 540$ ns (the pink shaded window in **b**). The gray dashed lines show equal spacing of 5.5 ± 0.2 MHz. The inset in (c) shows the acoustic traveling path within the gated time window. **(d)** The $|S_{21}|$ spectrum after time-gating from $190 < t < 350$ ns (the blue shaded window in **b**). The inset shows the acoustic wave traveling path within the gated time window.

[1] M. Bicer, S. Valle, J. Brown, M. Kuball, and K. C. Balram, “Gallium nitride phononic integrated circuits platform for GHz frequency acoustic wave devices,” *Appl. Phys. Lett.*, vol. 120, no. 243502, 2022.

[2] F. M. Mayor, W. Jiang, C. J. Sarabalis, T. P. McKenna, J. D. Witmer, and A. H. Safavi-Naeini, “Gigahertz Phononic Integrated Circuits on Thin-Film Lithium Niobate on Sapphire,” *Phys. Rev. Applied*, vol. 15, Jan 2021.

11, The authors simulate the mechanical and optical propagation in the device, and present a formula for calculating the optomechanical coupling g_{20} in equation 2. A full presentation of the physics of this device should contain simulations of this coupling strength from the photoelastic and moving boundary conditions they present in the first few equations in the supplementary information. What value do they simulate, and how does the value of g and $P_{\pi/2}$ they extract from their measurements compare with the simulated values?

Our response: We thank the reviewer for this suggestion. We have added the following discussion with simulation results in the revised Supplementary Information.

Here we discuss the simulation of the presented optomechanical system and calculate the optomechanical coupling coefficient g_0 . As stated above, the optomechanical coupling is facilitated through traveling waves in the presented OMR system, which is different from the previous standing-wave systems. Therefore, instead of simulating the whole OMR system, we simulate the acoustic and optical mode profile of a cross section of the waveguide, as shown in the main text Fig. 1. We use COMSOL 5.6 to simulate the cross-sectional mode profile and calculate g_0 from the moving boundary (MB) and the photoelastic (PE) contribution using the following expression

$$\frac{g_0}{\sqrt{\hbar\Omega}} = \left(\frac{G_{MB}}{\sqrt{P_a}} + \frac{G_{PE}}{\sqrt{P_a}} \right) \left(\frac{1}{\text{mm}\sqrt{W}} \right)$$

where the G_{MB} and G_{PE} are the optomechanical coupling coefficient contributed by the MB and PE effect, respectively. The following table lists the simulated value and the calculated g_0 . The calculated $g_0/\sqrt{\hbar\Omega} = 574.3 \text{ mm}^{-1}\sqrt{W}^{-1}$, which is about 2.5 times larger than the measured g_0 from data. The simulated $g_0/\sqrt{\hbar\Omega}$ has a $P_{\pi/2} = 0.02$ mW using the same Oby MR geometry as in the main text. Since the moving boundary effect dominates the coupling, we attribute the discrepancy in the $g_0/\sqrt{\hbar\Omega}$ to the refractive index difference between the simulated GaP and the BGaP used in the experiment.

$\omega_0/2\pi$ (THz)	$\Omega/2\pi$ (GHz)	PE contribution $G_{PE}/\sqrt{P_a} \left(\frac{1}{\text{mm}\sqrt{W}} \right)$	MB contribution $G_{MB}/\sqrt{P_a} \left(\frac{1}{\text{mm}\sqrt{W}} \right)$	Acoustic power P_a (W)
190.80	2.56	-7.9	582.2	3.8×10^{-3}

12, The coupled-mode fitting of the data requires a large number of fixed parameters. How are the values for γ_0 , γ_a , Γ and Γ_a for both the optics and the mechanical modes determined?

Our response: We revised the CMT model. The effect of γ_0 and Γ are now included in the TE₀ mode attenuation coefficient $a(P_a) = a_0 + b(P_a)$, where $a_0 = e^{-\alpha\pi D}$, with α being the propagation loss of the TE₀ mode, and $b(P_a) = A_0(z)/A_0(0)$ is the optomechanical coupling attenuation. The propagation loss α can be extracted by fitting the results in Fig. 2c. The acoustic wave in the OMR is treated with a similar model where a single acoustic attenuation coefficient b_a , which can be extracted by fitting the results in Fig. 2d. Here we include the updated Supplementary Table S1 using the revised CMT model.

Fixed parameters					Fit parameters	
$A_{0-in}(\sqrt{W})$	κ_0	κ_2	κ_a	b_a	$g/\sqrt{\hbar\Omega}$ ($\frac{1}{\text{mm}\sqrt{W}}$)	$\Delta\beta$ (μm^{-1})
0.02	0.95	0.17	0.80	0.86	230	0.079

13, As a route to increasing the device efficiency, the authors suggest thermal tuning of the phase-mismatch, however the device needs to be operated at cryogenic temperatures. How would they realise zero phase mismatch in these conditions?

Our response: The phase mismatch in our system is due to the deviation of the OMR's optical resonance frequency from the designed value. Thermal tuning of the optical resonance can be achieved by using a local, micro-sized electrical heater integrated near the OMR [1]. On the other hand, the acoustic resonance frequency is less sensitive to thermal tuning, compared with optical resonance. Although the devices are operated at cryogenic temperature, local thermal heating can be effective, consuming less than μW of power [2][3]. With careful design, the local heater could tune the optical resonance with a minimal effect on the overall device temperature.

[1] Alan D. Logan et al, "Triply-resonant sum frequency conversion with gallium phosphide ring resonators," *Opt. Express* 31, 1516-1531 (2023)

[2] R. Amatya, C. W. Holzwarth, H. I. Smith and R. J. Ram, "Precision Tunable Silicon Compatible Microring Filters," in *IEEE Photonics Technology Letters*, vol. 20, no. 20, pp. 1739-1741, Oct.15, 2008, doi: 10.1109/LPT.2008.2004680.

[3] Elshaari, Ali W., et al. "Thermo-optic characterization of silicon nitride resonators for cryogenic photonic circuits." *IEEE Photonics Journal* 8.3 (2016): 1-9. <https://doi.org/10.1109/JPHOT.2016.2561622>

14, In the conclusion the authors state that 'we have demonstrated the first OMR in which photonic and phononic modes are co-resonantly coupled to achieve efficient optomechanical mode conversion with a record low critical acoustic power ($P\pi/2$).' This is not the case from what they have presented - they have projected the number they claim to be record low and not demonstrated it. This needs to be corrected.

Our response: We agree and have revised the claim in the conclusion as follows.

In conclusion, we have demonstrated the first micro-scale OMR in which photonic and phononic modes are co-resonantly coupled to achieve efficient optomechanical mode conversion.

15, A lot of details remain missing about the potential quantum applications for an improved device, something quoted as a key motivation for this work. The authors should be much more specific as to the actual form of conversion they want to pursue with this device. They discuss the potential for ultra-efficient conversion from microwave to optical signals, however they use very large mechanical powers containing macroscopic numbers of microwave photons. They need to break down what the efficiency of a single-photon-level microwave signal conversion would be for this device in order to validate their outlook. What optical power would be required for efficient microwave-to-optical conversion?

Our response: Although the ultimate goal of the systems is for quantum transduction, our current system is still far from that regime, as the reviewer points out. Nevertheless, our work presents the first demonstration of co-resonating traveling photons and phonons in an optomechanical ring resonator, currently operated in the classical regime.

In the revised Supplementary Information, we have included the following discussion of the microwave-to-optical transduction efficiency and the optical power required for quantum-level conversion.

FIG. S3. General schematic for microwave-to-optical conversion.

A general schematic of the microwave-to-optical conversion of our device is shown in Fig. S3. Here, the microwave-to-optical conversion is a two-stage conversion, where the microwave signal is coupled to the mechanical resonator via electromechanical coupling with coupling rate g_{EM} , and the mechanical resonator is coupled to the optical resonator via optomechanical coupling with coupling rate g_{OM} . In this section, we will characterize key parameters related to the microwave-to-optical transduction. The following calculations are based on the work of Wu et al. [1], Han et al. [2] and Aspelmeyer et al. [3].

A. Calculating Cooperativities

A general form of the cooperativity is expressed as

$$C_{ij} = \frac{4g_{ij}^2}{k_i k_j}$$

where $i, j = (o, m, e)$ for two-stage microwave-to-optical transduction, namely, electromechanical (EM) and optomechanical (OM) coupling. And $k_i (k_j)$ is the mechanical damping rate of the i^{th} (j^{th}) stage. C_{EM} is

the cooperativity between the microwave and the mechanical resonator, and the C_{OM} is the cooperativity between the optical and the mechanical resonator.

1. Calculating C_{EM}

First, we calculate the C_{EM} using the expression

$$C_{EM} = \frac{4g_{EM}^2}{\gamma_m k_e}$$

where g_{EM} is the electromechanical coupling rate, γ_m is the mechanical energy loss rate (unit: $\text{Hz} \cdot 2\pi$), and k_e is the electrical decay rate (FWHM). The g_{EM} (unit: $\text{Hz} \cdot 2\pi$) can be extracted from the modeled effective circuit in [3] using

$$g_{EM} = \frac{\sqrt{k_T^2} \omega_m}{2}$$

$$k_T^2 = \frac{C_m}{C_m + C_p + C_T}$$

where k_T^2 is the reduced piezoelectric coupling strength and $\omega_m = \omega_{LC}$ is the resonance frequency of the RF reflection spectrum. In our model, C_T is absorbed into C_p since the two capacitors are connected in parallel. The modified Butterworth Van Dyke (mBVD) model of the piezoelectric circuit for our system is shown in Fig. S4(a). The electrical decay rate k_e can be extracted from the FWHM of the RF reflection spectrum at resonance frequency, which is shown in Fig. S4(c).

We use the mBVD model to fit the S_{11} data to extract C_p and C_m . Fig. S4(c)(d) show the fitting result of the mBVD model. The fitted $C_p = 0.82 \text{ pF}$ and $C_m = 0.0013 \text{ pF}$. The fitting result is consistent with the measured data, and we can use the extracted capacitance value to calculate

$$k_T^2 = \frac{C_m}{C_m + C_p + 0} = \frac{0.0013 \text{ pF}}{0.82 \text{ pF} + 0.0013 \text{ pF}} \approx 1.582 \times 10^{-3}$$

$$\frac{\omega_m}{2\pi} = 2.556 \text{ (GHz)}$$

$$g_{EM} = \frac{\sqrt{k_T^2} \omega_m}{2} = 319.5 \left(\frac{\text{Mrad}}{\text{s}} \right)$$

$$k_e = 15 \text{ (MHz)}$$

$$\gamma_m = 1.1 \text{ (MHz)}$$

and we can calculate the C_{EM} as

$$C_{EM} = \frac{4g_{EM}^2}{\gamma_m k_e} = \frac{4 \times (319.5 \times 10^6)^2}{15 \times 10^6 \times 1.1 \times 10^6 (2\pi)^2} = 626$$

2. Calculating C_{OM}

Optomechanical cooperativity can be calculated as:

$$C_{OM} = \frac{4g_{OM}^2}{\gamma_m \kappa_o}$$

where g_{OM} is the pump-enhanced coupling rate $g_{OM} = g_0\sqrt{n_{phot}}$, and n_{phot} is the intracavity photon number. Single photon optomechanical coupling rate g_0 can be simulated using finite element analysis (FEA). Experimentally, g_0 can be extracted from the measured data and $\sqrt{n_{phot}}$ can be calculated from the optical pump power. γ_m can be extracted from the acoustic quality factor, and κ_o can be extracted from the optical quality factor. The phonon-flux normalized optomechanical coupling rate $g_{extract}/\sqrt{\hbar\Omega} = 230 \text{ (mm}^{-1}\sqrt{W}^{-1}\text{)}$ and we can calculate the pump-enhanced coupling rate as

$$g_{OM} = g_0\sqrt{N_{phot}} = \left(\frac{g_{extract}}{\sqrt{\hbar\Omega}} \sqrt{P_a} \times \pi D \right) \times \sqrt{N_{phot}}$$

where D is the diameter of the OM ring and P_a is the RF pump power and Ω is the frequency of the L_2 mode. We first perform the following calculations.

$$\begin{aligned} g_0\sqrt{N_{phot}} &= \frac{g_{extract}}{\sqrt{\hbar\Omega}} \sqrt{1.5 \times 10^{-3} \times 200 \times 10^{-3} \pi \sqrt{N_{phot}}} \\ &= 5.7\sqrt{N_{phot}} = 293 \times 10^6 = g_{OM} \\ \gamma_m &= 1.1 \text{ MHz} \\ \kappa_o &= 2.43 \text{ GHz} \end{aligned}$$

And we can calculate C_{OM} as

$$C_{OM} = \frac{4g_{OM}^2}{\gamma_m \kappa_o} = \frac{4 \times (293 \times 10^6)^2}{1.1 \times 10^6 \times 2.43 \times 10^9 (2\pi)^2} = 3.35$$

- [1] M. Wu, E. Zeuthen, K. C. Balram, and K. Srinivasan, “Microwave-to-optical transduction using a mechanical supermode for coupling piezoelectric and optomechanical resonators,” *Phys. Rev. Appl.*, vol. 13, p. 014027, Jan 2020.
- [2] X. Han, W. Fu, C.-L. Zou, L. Jiang, and H. X. Tang, “Microwave-optical quantum frequency conversion,” *Optica*, vol. 8, pp. 1050–1064, Aug 2021.
- [3] M. Aspelmeyer, T. J. Kippenberg, and F. Marquardt, “Cavity optomechanics,” *Reviews of Modern Physics*, vol. 86, pp. 1391–1452, dec 2014.

16. Additionally - there is no mention of the noise that might be added during conversion. Discussion of these details need to be contained in an outlook concerning the quantum applications of a device.

Our response: Similar to the previous question, we have included a discussion of the added noise of our system in the revised Supplementary Information, which is also included below for your reference.

B. Added noise

Following the treatment in reference [1], the added noise N arises from two main contributions in our platform: optical noise contributed by the Stokes scattering N_o and the thermomechanical noise from the pumped phonon N_m . The optical added noise can be expressed as

$$N_o = \frac{1}{\eta_e} \frac{C_{OM} \mathcal{L}^2}{C_{EM}}$$

where the electrical coupling efficiency is given by $\eta_e = \frac{Z_{tx}}{Z_{tx} + R_s}$. Z_{tx} is the characteristic impedance and R_s is the serial resistance of the mBVD model. The optical-cavity Lorentzian sideband amplitudes are expressed as

$$\mathcal{L}_{\pm}^2 = \frac{\left(\frac{\kappa_o}{2}\right)^2}{\left(\frac{\kappa_o}{2}\right)^2 + (\Delta \pm \omega_m)^2}$$

where $\Delta = \omega_{pump} - \omega_0$ is the laser detuning from cavity resonance, κ_0 is the optical mode decay rate, ω_m is the acoustic resonance frequency. In our device, the calculated Stokes sideband amplitude $\mathcal{L}_{-}^2 = 0.18$, and we can use the information calculated so far to calculate optical added noise as

$$N_o = \frac{1}{0.72} \frac{(3.35 \times 0.18)}{626} = 0.0013$$

The mechanical thermal noise can be expressed as

$$N_m = \frac{1}{\eta_e} \frac{n_m}{C_{EM}} = \frac{1}{0.72} \frac{33}{626} = 0.07$$

where $n_m(\omega) = [e^{\hbar\omega/(k_B T)} - 1]^{-1}$ is given by the Bose-Einstein distribution and $\omega = \omega_m$. And the total added noise is the sum of the two contributions: $N_{add} = N_o + N_m = 0.0713$.

A couple of additional minor formatting notes.

1, In the supplementary the coupled mode theory uses dimensionless parameter for the coupling in and out of the resonator. These numbers are consistently referred to as ‘coupling rate’ which is highly misleading, as they are dimensionless transmissivities rather than rates.

Our response: We thank the reviewer for spotting this error. Those dimensionless parameters should be named “coupling coefficient” or “cross-coupling coefficient”. We have corrected this in the Supplementary Information.

2, There seems to be some inconsistency in the labelling of the ‘through port’ the optical input and through are labelled clearly in figure 2a, however I believe the model of the OMR presented in the equation between equations 4 and 5 of section I of the supplementary refers to the coupling to the drop port, rather than the through port.

Our response: We thank the reviewer for pointing this issue out. Indeed, A_{out} in equation (5) in the Supplementary Information refers to the drop port output. To clarify, we have changed it to A_{drop} in the revised Supplementary Information, as defined in the figure below.

FIG. S1. CMT model of the OMR

REVIEWER COMMENTS

Reviewer #1 (Remarks to the Author):

I have read the revised manuscript along with the report of the other referee as well as the authors' response. In my view the work is valid and the manuscript is suitable for publication after a few minor clarifications.

Most concerns raised during the first review have been addressed by the authors. However, there is still some confusion regarding the achieved conversion efficiency. In the initial abstract, the authors mentioned a conversion efficiency of 8.2% at 1.6mW. Now it is modified to 'internal conversion efficiency $2.1 \cdot 10^{-2}$ ' and 'total conversion efficiency $0.57 \cdot 10^{-6}$ '. In the revised manuscript they now mention the highest internal conversion efficiency $\eta_i = 2.1 \pm 0.1$ at $P_a = 1.6$ mW, with NO mention of 10^{-2} .

Could the authors please clarify whether the ' 10^{-2} ' is a typo and explain why the reported efficiency differs from the original 8.2% value in the abstract? In a brief statement, it would be helpful if the authors could define what exactly conversion efficiency is achieved.

Finally, I don't think that the internal efficiency ONLY depends on phase mismatch. If present in the main text, this statement should be removed.

Reviewer #2 (Remarks to the Author):

I am satisfied with the revisions that the authors have made in response to my queries. I now support publication of this manuscript.

The Authors' Response to Reviewers' Comments

Reviewer #1 (Remarks to the Author):

I have read the revised manuscript along with the report of the other referee as well as the authors' response. In my view the work is valid and the manuscript is suitable for publication after a few minor clarifications.

Our response: We thank the reviewer for the comments on our work and the support for publication.

Most concerns raised during the first review have been addressed by the authors. However, there is still some confusion regarding the achieved conversion efficiency. In the initial abstract, the authors mentioned a conversion efficiency of 8.2% at 1.6mW. Now it is modified to 'internal conversion efficiency 2.1×10^{-2} ' and 'total conversion efficiency 0.57×10^{-6} '. In the revised manuscript they now mention the highest internal conversion efficiency $\eta_i = 2.1 \pm 0.1$ at $P_a = 1.6$ mW, with NO mention of 10^{-2} .

Could the authors please clarify whether the ' 10^{-2} ' is a typo and explain why the reported efficiency differs from the original 8.2% value in the abstract? In a brief statement, it would be helpful if the authors could define what exactly conversion efficiency is achieved.

Our response:

A pair of parentheses should be added in the expression of the internal conversion efficiency in the main text such that it reads $(2.1 \pm 0.1)\%$. We have corrected it and changed “%” to scientific notation to be consistent with the expression in the abstract.

“the highest internal conversion efficiency $\eta_i \equiv \frac{P_{TE_2}}{P_{TE_0}} = (2.1 \pm 0.1) \times 10^{-2}$ at $P_a = P_{\pi/2} = 1.6$ mW”.

Furthermore, based on the reviewers' comments in the previous round of review, in the previous revision, we have updated our calculation of the internal conversion efficiency with very detailed analysis in the response and the supplementary materials.

Finally, I don't think that the internal efficiency ONLY depends on phase mismatch. If present in the main text, this statement should be removed.

Our response: We agree with the reviewer that phase mismatch is not the ONLY factor affecting internal efficiency, as other factors such as loss also play a role. We have revised the main text as the following.

The full parameter set that is used for fitting is listed in Supplementary Table S1. From the fitting, we extract the coupling coefficient $\frac{g}{\sqrt{\hbar\Omega}} = 230 \text{ mm}^{-1}\sqrt{\text{W}^{-1}}$ and the phase-mismatch $\Delta\beta = 0.08 \mu\text{m}^{-1}$, respectively. ~~Because of phase mismatch,~~ The achieved highest internal conversion efficiency $\eta_i \equiv \frac{P_{TE_2}}{P_{TE_0}} = (2.1 \pm 0.1) \times 10^{-2}$ at $P_a = 1.6$ mW, which agrees with the theory (See Supplementary Information II). The phase-mismatch can be mitigated by tuning the resonance frequency of OMR using photothermal tuning⁶⁴. We can calculate the critical acoustic power needed to achieve unity conversion efficiency in the OMR at phase matching condition ~~is achieved~~:

$$P_{\pi/2} = \left(\frac{\sqrt{\hbar\Omega}}{2gD} \right)^2 = 0.1 \text{ mW}$$

where $D = 200 \mu m$ is the diameter of the OMR. The $P_{\pi/2} = 0.1 mW$ is projected at phase matching condition for unity conversion efficiency, however, in our experimental we can only achieve $P_{\pi/2} = 1.6 mW$ with a total conversion efficiency $\eta_{tot} \equiv \frac{P_{out}}{P_{laser}} = 0.57 \times 10^{-6}$.

Reviewer #2 (Remarks to the Author):

I am satisfied with the revisions that the authors have made in response to my queries. I now support publication of this manuscript.

Our response: We thank the reviewer for supporting the publication of our work.

REVIEWERS' COMMENTS

Reviewer #1 (Remarks to the Author):

I am satisfied with the authors' response. And I recommend the manuscript for publication.

The Authors' Response to Reviewers' Comments

REVIEWERS' COMMENTS

Reviewer #1 (Remarks to the Author):

I am satisfied with the authors' response. And I recommend the manuscript for publication.

Our response: We thank the reviewer for spending time reviewing our manuscript and supporting its publication.